# Ovipositor and mouthparts in a fossil insect support a novel ecological role for early orthopterans in 300 million years old forests

Lu Chen[1], Jun-Jie Gu[2], Qiang Yang[3], Dong Ren[1]*, Alexander Blanke[4†], Olivier Béthoux[5†]

[1]College of Life Sciences and Academy for Multidisciplinary Studies, Capital Normal University, Beijing, China; [2]Institute of Ecological Agriculture, College of Agronomy, Sichuan Agricultural University, Chengdu, China; [3]School of Life Sciences, Guangzhou University, 230 Waihuanxi Road, Guangzhou Higher Education Mega Center, Guangzhou, China; [4]Institute of Evolutionary Biology and Animal Ecology, University of Bonn, Bonn, Germany; [5]CR2P (Centre de Recherche en Paléontologie – Paris), MNHN – CNRS – Sorbonne Université; Muséum National d'Histoire Naturelle, Paris, France

**Abstract** A high portion of the earliest known insect fauna is composed of the so-called 'lobeattid insects', whose systematic affinities and role as foliage feeders remain debated. We investigated hundreds of samples of a new lobeattid species from the Xiaheyan locality using a combination of photographic techniques, including reflectance transforming imaging, geometric morphometrics, and biomechanics to document its morphology, and infer its phylogenetic position and ecological role. *Ctenoptilus frequens* sp. nov. possessed a sword-shaped ovipositor with valves interlocked by two ball-and-socket mechanisms, lacked jumping hind-legs, and certain wing venation features. This combination of characters unambiguously supports lobeattids as stem relatives of all living Orthoptera (crickets, grasshoppers, katydids). Given the herein presented and other remains, it follows that this group experienced an early diversification and, additionally, occurred in high individual numbers. The ovipositor shape indicates that ground was the preferred substrate for eggs. Visible mouthparts made it possible to assess the efficiency of the mandibular food uptake system in comparison to a wide array of extant species. The new species was likely omnivorous which explains the paucity of external damage on contemporaneous plant foliage.

**\*For correspondence:**
rendong@mail.cnu.edu.cn

[†]These authors share joint senior authorship to this work

**Competing interest:** The authors declare that no competing interests exist.

## Introduction

The earliest known insect fauna in the Pennsylvanian, ca. 307 million years ago, was composed by species displaying mixtures of inherited (plesiomorphic) and derived (apomorphic) conditions, such as the griffenflies (stem relatives of dragon- and damselflies), but also by highly specialized groups, such as the gracile and sap-feeding megasecopterans, belonging to the extinct taxon Rostropalaeoptera. A prominent portion of this fauna were the so-called 'lobeattid insects'. They have been recovered from all major Pennsylvanian outcrops, where some species can abound (*Béthoux, 2005c*; *Béthoux, 2008*; *Béthoux and Nel, 2005a*). Indeed, at the Xiaheyan locality, China, for which quantitative data are available, they collectively account for more than half of all insect occurrences (*Trümper et al., 2020*). Additionally, another extinct group, the Cnemidolestodea, composed of derived relatives of

lobeattid insects, was likewise ubiquitously distributed during the Pennsylvanian until the onset of the Permian (*Béthoux, 2005b*).

The phylogenetic affinities of lobeattid insects are debated. They have been regarded as stem relatives of either Orthoptera (crickets, grasshoppers, katydids; *Béthoux and Nel, 2002*; *Béthoux and Nel, 2005a*) or of several other lineages within the diverse Polyneoptera (*Aristov, 2014*; *Rasnitsyn, 2007*). A core point of the debate is the presumed wing venation ground pattern of insects, which, however, will remain elusive until Mississipian or even earlier fossil wings are discovered. Ecological preferences of lobeattid insects are also poorly known. Traditionally, they have been regarded as foliage feeders (*Labandeira, 1998*) but, given their abundance, this is in contrast to the paucity of documented external foliage damage during that time.

The Xiaheyan locality is unique in several respects (*Trümper et al., 2020*), including the amount of insect material it contains. Over the past decade, a collection of several thousand specimens was unearthed, allowing for highly detailed analyses of, for example, ovipositor and mouthparts morphology of extinct insect lineages (*Pecharová et al., 2015b*). These character systems are investigated herein in a new lobeattid species, based on hundreds of remains, using reflectance transforming imaging (RTI) together with more traditional approaches. Dietary preferences were inferred using a comparative morphometric and biomechanical analysis of gnathal edge shape based on an extensive dataset of extant polyneopteran species, with a focus on Orthoptera. Together, this investigation provides information regarding the phylogenetic affinities of loebattid insects and on their preferred mode of egg laying and dietary niche.

## Results
### Systematic palaeontology

> Archaeorthoptera *Béthoux and Nel, 2002*
> Ctenoptilidae *Aristov, 2014*
> Ctenoptilus *Lameere, 1917*
> *Ctenoptilus frequens Chen et al., 2020*
> LSID (Life Science Identifier). F0D67EC6-1C1A-4A8E-A8C0-31641FD057E3

### Etymology
Based on '*frequens*' ('frequent' in Latin), referring to the abundance of the species at the Xiaheyan locality. Holotype. Specimen CNU-NX1-326 (female individual; *Figure 1*).

Referred material. See Appendix 1, Section 2.1.2.

### Locality and horizon
Xiaheyan Village, Zhongwei City, Yanghugou Formation (Ningxia Hui Autonomous Region, China); latest Bashkirian (latest Duckmantian) to middle Moscovian (Bolsovian), early Pennsylvanian (*Trümper et al., 2020*).

### Differential diagnosis
The species is largely similar to *Ctenoptilus elongatus* (*Brongniart, 1893*), in particular in its wing venation (Appendix 1, Section 2.1.2). However, it differs from it in its smaller size (deduced from forewing length) and its prothorax longer than wide (as opposed to quadrangular).

General description. See Appendix 1, Section 2.1.2.

### Specimens description
See Appendix 1, Section 2.1 and *Appendix 1—figures 2–8*; details of ovipositor, see *Figure 2*; details of head, see Figure 4.

## Ovipositor morphology
The external genitalia in insects consist primarily of a pair of mesal extensions, the so-called gonopods, or ovipositor blades, and a pair of lateral projections, the so-called gonostyli, or ovipositor sheaths on abdominal segments 8 and 9. These sclerotized elements are collectively referred to as 'valves'. The

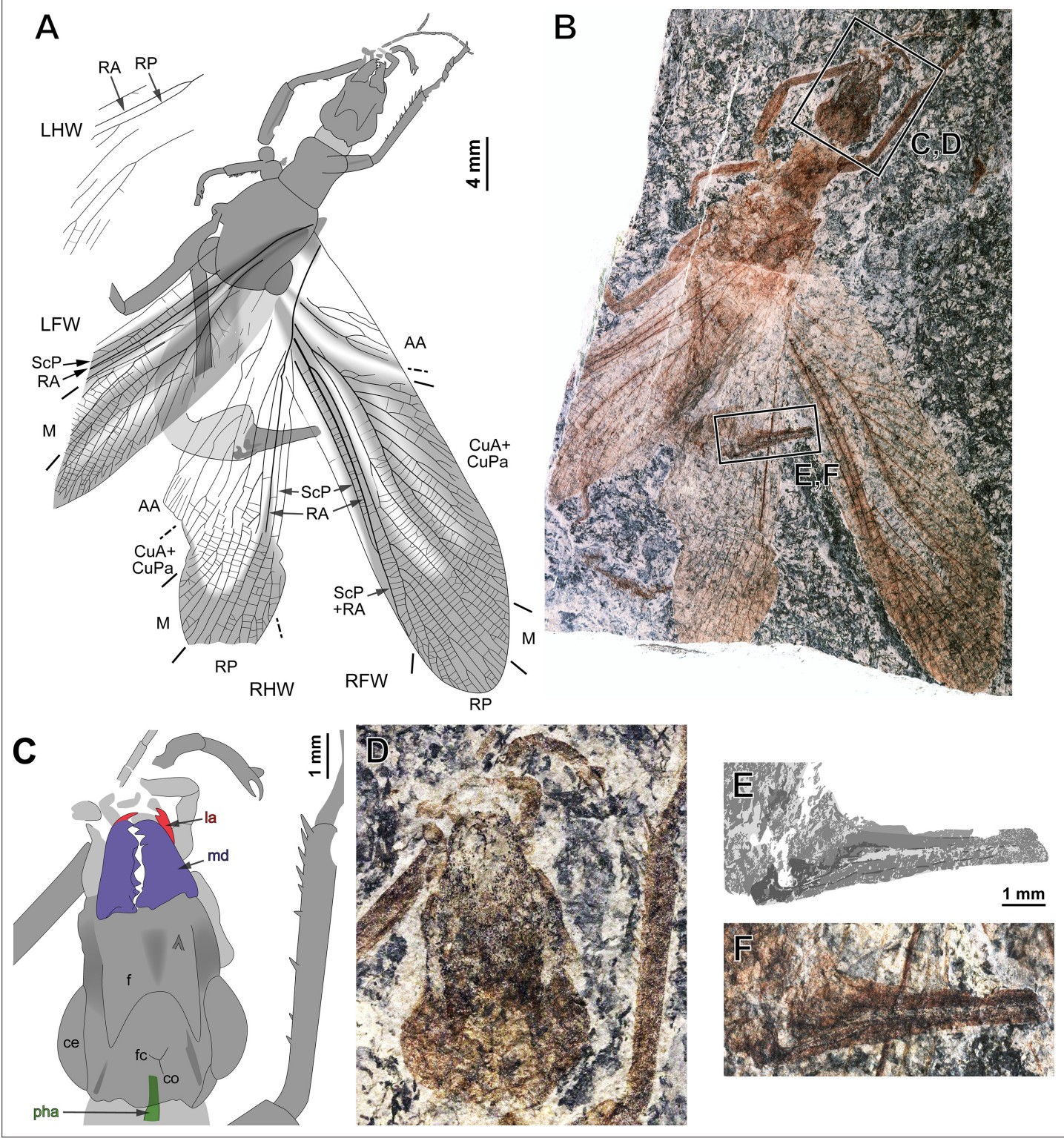

**Figure 1.** *Ctenoptilus frequens* sp. nov., holotype (CNU-NX1-326). (**A**) Habitus drawing and (**B**) habitus photograph (composite); (**C–D**) details of head and right foreleg (location as indicated in **B**), (**C**) color-coded interpretative drawing and (**D**) photograph (composite); and (**E–F**) details of ovipositor (location as indicated in **B**), (**E**) drawing and (**F**) photograph (composite). Color-coding and associated abbreviations: red, lacina (la); dark blue-purple, mandible (md); green, pharynx (pha). Other indications, head: ce, composite eye; f, frons; co, coronal cleavage line; fc, frontal cleavage line. Wing morphology abbreviations: LFW, left forewing; LHW, left hind wing; RFW, right forewing; RHW, right hind wing; ScP, posterior subcosta; RA, anterior radius; RP, posterior radius; M, media; CuA, anterior cubitus; CuPa, anterior branch of posterior cubitus; CuPb, posterior branch of posterior cubitus; AA,

*Figure 1 continued on next page*

*Figure 1 continued*

anterior analis. Photograph (composite). Color-coding and associated abbreviations: red, lacina (la); dark blue-purple, mandible (md); green, pharynx (pha). Other indications, head: ce, composite eye; f, frons; co, coronal cleavage line; fc, frontal cleavage line. Wing morphology abbreviations: LFW, left forewing; LHW, left hind wing; RFW, right forewing; RHW, right hind wing; ScP, posterior subcosta; RA, anterior radius; RP, posterior radius; M, media; CuA, anterior cubitus; CuPa, anterior branch of posterior cubitus; CuPb, posterior branch of posterior cubitus; AA, anterior analis.

studied fossils possess three pairs of valves in their ovipositor, each strongly sclerotized (*Figure 2*, and *Appendix 1—figure 7B, C, 8B and C*). Especially the valve margins are still visible in the anterior area ('base'), including the dorsal margin of the gonostylus IX (gs9), the ventral margin of the gonapophysis IX (gp9), and the dorsal and ventral margins of gonapophysis VIII (gp8). All observed ovipositors, but in particular the one of specimen CNU-NX1-742 (*Figure 2D–F*, and *Appendix 1—figure 8B and*

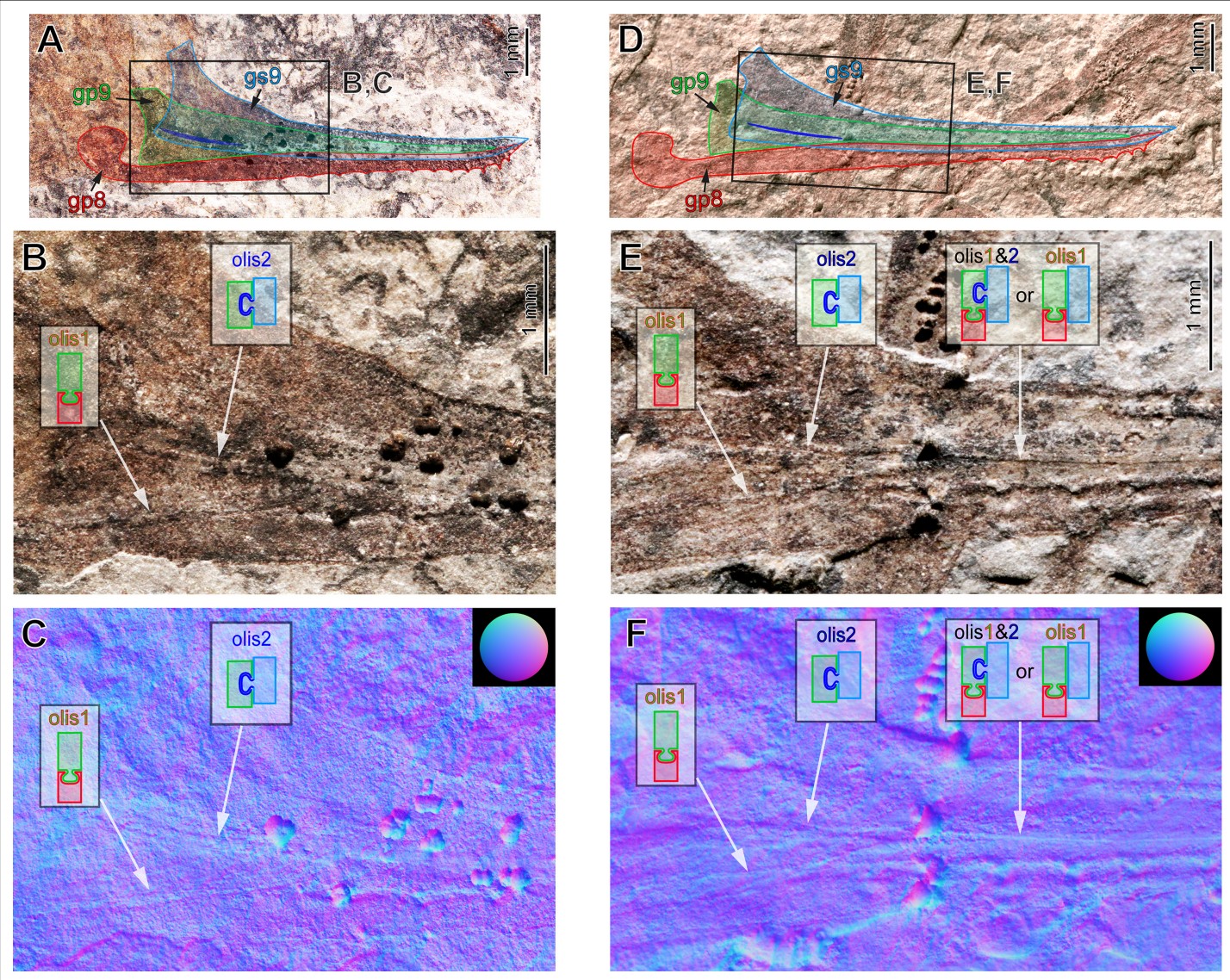

**Figure 2.** External ovipositor in *Ctenoptilus frequens* sp. nov. in lateral view. (**A–C**) Specimen CNU-NX1-749, (**A**) overview of the ovipositor with overlaid indications of the ovipositor parts (see also) overview of the ovipositor with overlaid indications of the ovipositor parts (see also *Appendix 1—figure 7A–C*) and (**B, C**) details of basal part of the same ovipositor as in **A**. (**B**) composite photograph and (**C**) reflectance transforming imaging (RTI) extract in normals visualization; (**D–F**) specimen CNU-NX1-742, (**D**) overview of the ovipositor with overlaid indications of the ovipositor parts (see also) overview of the ovipositor with overlaid indications of the ovipositor parts (see also *Appendix 1—figure 8B and C*) and (**E, F**) details of basal part of the same ovipositor as in **D**; (**E**) composite photograph and (**F**) RTI extract in normals visualization. Olistheter ('olis') configurations at different parts of each respective ovipositor are shown as insets. Abbreviations: Gonostylus IX (gs9); gonapophysis IX (gp9); gonapophysis VIII (gp8).

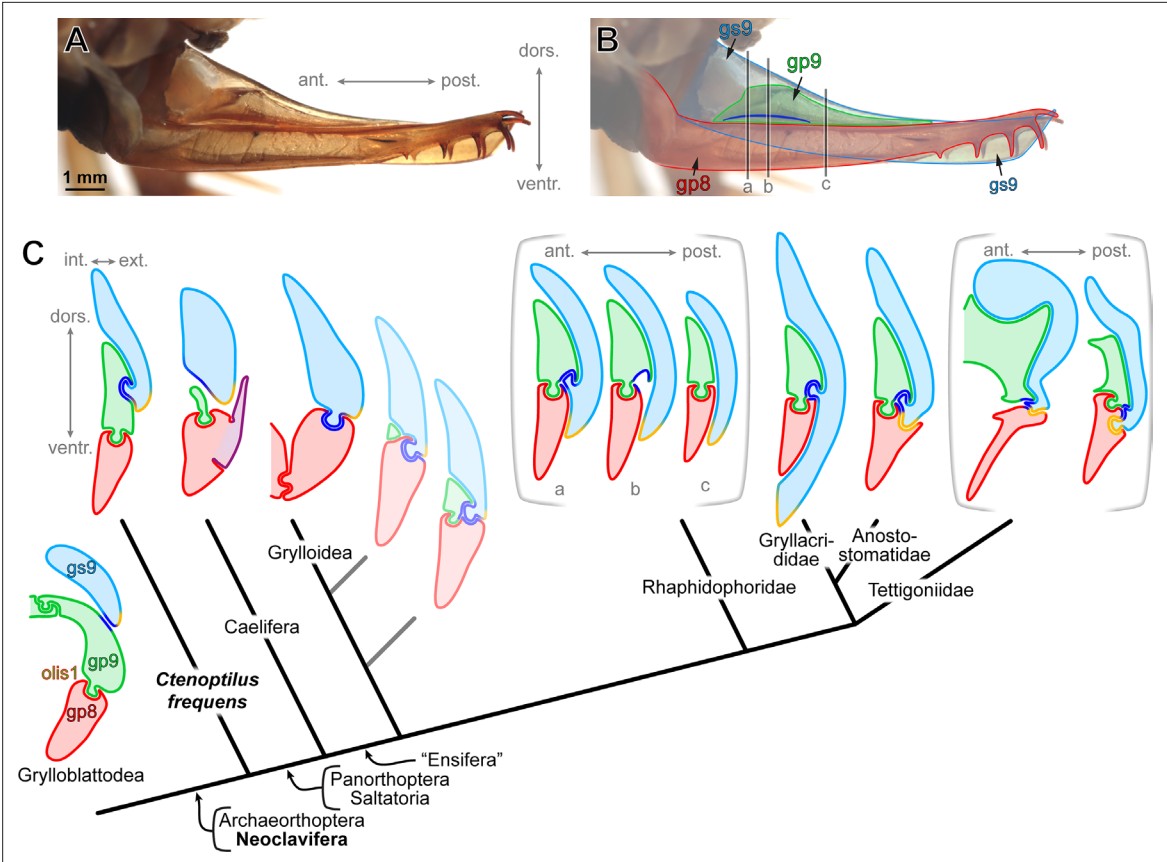

**Figure 3.** The evolution of major ovipositor configurations across Orthoptera. (**A**) External ovipositor of external ovipositor of *Ceuthophilus* sp. (Orthoptera: Rhaphidophoridae; extant species) in laterial view (left side, flipped horizontally, left gonostylus IX [gs9] removed). (**B**) Same as above, but annotated. The three black vertical lines labelled 'a', 'b', 'c' indicate the position of the three schematic sections shown in **C**. (**C**) Schematic ovipositor cross-sections in Grylloblattodea, *Ctenoptilus frequens* sp. nov., and several extant Orthoptera possessing well-developed ovipositors (not to scale; (see Appendix 1, Section 2.2). Ovipositor configurations are mapped onto the phylogenomic inference carried out by ***Song et al., 2020***. Pale cross-section along the stem of Grylloidea is hypothetical; sections delineated by brackets represent conditions along the antero-posterior axis. Color-coding and associated abbreviations: light blue, gonostylus IX (gs9; light green, gonapophysis IX (gp9); red, gonapophysis VIII (gp8); royal blue, secondary olistheter (olis2); light orange, tertiary olistheter (olis3); purple, 'lateral basivalvular sclerite' (specific to Caelifera). Other indications: olis1, primary olistheter; int./ ext., internal/external, respectively; dors./ventr., dorsal/ventral, respectively; and ant./post., anterior/posterior, respectively.

**C**), display, from the second third of their length onwards, a thin longitudinal line much sharper and more developed than other visible linear structures in the area. This is the primary olistheter (olis1), a tongue-like structure which commonly interlocks gp9 and gp8 in extant insects having more or less well-developed external ovipositors (***Figure 3***; ***Klass, 2008***). In the distal half of the ovipositor, the linear structure occurring between the dorsal edge of gs9 and olis1 is interpreted as the dorsal margin of gp9.

Together with the position of the antero-basal apophysis (=outgrowth) of this valve, the anterior margin of gp9 can then be traced. The extent of olis1 indicates that gp9 reaches the ovipositor apex, which is corroborated by the length of its inferred dorsal margin, well visible in specimen CNU-NX1-749 (***Figure 2A***, and ***Appendix 1—figure 7B and C***). This specimen also shows that gp8 bears ventrally oriented teeth, more prominent and densely distributed near the apex, as in many extant orthopterans. The location of the dorsal margin of gp9 could not be observed with confidence near the base, which might be due to a lower degree of sclerotization.

This morphology implies that, at the base, dorsal to the anterior margin of gp8, only gs9 and gp9 occur. Therefore, the sharp and heavily sclerotized longitudinal line, located slightly dorsally with respect to the ventral margin of gs9, can only be an olistheter interlocking these two valves. This second olistheter (olis2) reaches olis1 but its development beyond this point could not be inferred with the available material. The occurrence of a mechanism locking gs9 onto gp9 is further supported

by the fact that these valves remained connected to each other in the specimen CNU-NX1-742 even though it endured heavy decay (head and ovipositor detached from thorax and abdomen, respectively; *Appendix 1—figure 8*).

## Mandibular mechanical advantage

The head and mouthpart morphology could be investigated in more detail in six specimens (see Appendix 1) while we could study the mechanical advantage (MA; see Section 1.5 of Appendix 1) of their mandibles in four of the six (viz. CNU-NX1-326, −747,−754, −764). The MA is defined as the inlever to outlever ratio and thus indicates the percentage of force transmitted to the food item (i.e. the effectivity of the lever system). Therefore, the MA allows for a size-independent comparison of the relative efficiencies of force transmission to the food item. Low MA values usually indicate quick biting with low force transmission typical for predators, while high MA values indicate comparatively slow biting with higher force transmission typical for non-predatory species.

Calculation of the MA along the entire gnathal edge revealed characteristic MA curve progressions for the studied taxa (Appendix 1, Section 2.3, and *Appendix 1—figure 9*). Compared to the studied fossils, extant Dermaptera, Embioptera, and Phasmatodea showed comparatively high MAs with an almost linear curve progression towards more distal parts of the mandibular incisivi whereas Plecoptera, Zoraptera, and Grylloblattodea were located at the lower end of the MA range with a gently exponential decrease towards the distal incisivi. The analysed extant Orthoptera occupy a comparatively wide functional space, with lineages at the higher and lower ends of the MA range. The composite fossil mandible representation (CFMR) of *Ct. frequens* (see Materials and methods) is located in the centre of the observed range of MAs for Orthoptera (*Figure 4*).

A polynomial function of the fifth order resulted in the best relative fit on the MA curves according to the Akaike information criterion (AIC) value (−661.3, see Materials and methods). The five common coefficients were subjected to a principal component analysis (PCA, *Figure 4E*), and, because phylogenetic signal was detected (K = 1.03316; p = 0.0001), also analysed using a phylogenetic principal component analysis (pPCA) (Appendix 1, Section 2.3, and *Appendix 1—figure 10*). The first four principal components (PCs) accounted for 96.8 % (PCA)/96 % (pPCA) of the variation in MA (*Appendix 1—table 2*).

In both PCAs, PC1 mainly codes for the vertical position of the MA curve, that is, the effectivity of the force transmission along the whole toothrow, while PC2 mainly codes for the curvature, that is, whether there is an almost linear or a gently exponential decrease in the effectivity of force transmission. Due to the narrow distribution of species along PC3, it was not possible to associate a clear biomechanical pattern to this PC.

The CFMR of *Ct. frequens* is located at the centre of the first three PCs (*Figure 4E*). Omnivorous Orthoptera and all herbivore taxa, with the exception of *Apotrechus*, are located along the width of PC1, while there is a tendency for the carnivorous taxa within the sampling to be spread along PC2.

## Discussion

### Phylogenetic implications

Our analysis of material of *Ct. frequens* provides unequivocal evidence that olis2 occurs in this species. Therefore, the new species was an orthopteran. The ovipositor configuration in *Ct. elongatus* furthermore conforms that observed in extant cave crickets (Raphidophoridae) in which olis2 occurs in addition to olis1 and interlocks gs9 and gp9 (*Figure 3A–C*; Appendix 1, Section 2.2). Indeed, this structure is present in ensiferan ('sword-bearing') Orthoptera possessing a developed ovipositor and is absent in caeliferan ('chisel-bearing') Orthoptera (*Cappe de Baillon, 1920*; *Cappe de Baillon, 1922*; *Kluge, 2016*; and see below). It follows that the new species is either more closely related to Ensifera than to Caelifera (owing to the possession of olis2), or it is a stem-orthopteran and olis2 was secondarily lost in Caelifera.

Further evidence for the phylogenetic placement of *Ct. frequens* is based on the lack of jump-related specializations in the hind-leg. Such specializations define the taxon Saltatoria within Orthoptera, and therefore *Ct. frequens* can be confidently excluded from crown-Orthoptera. This conclusion is furthermore corroborated by wing vein characteristics: *Ct. frequens* lacked a forked CuPa vein before its fusion with the CuA vein. Such a forked CuPa vein is typical for Panorthoptera, which includes

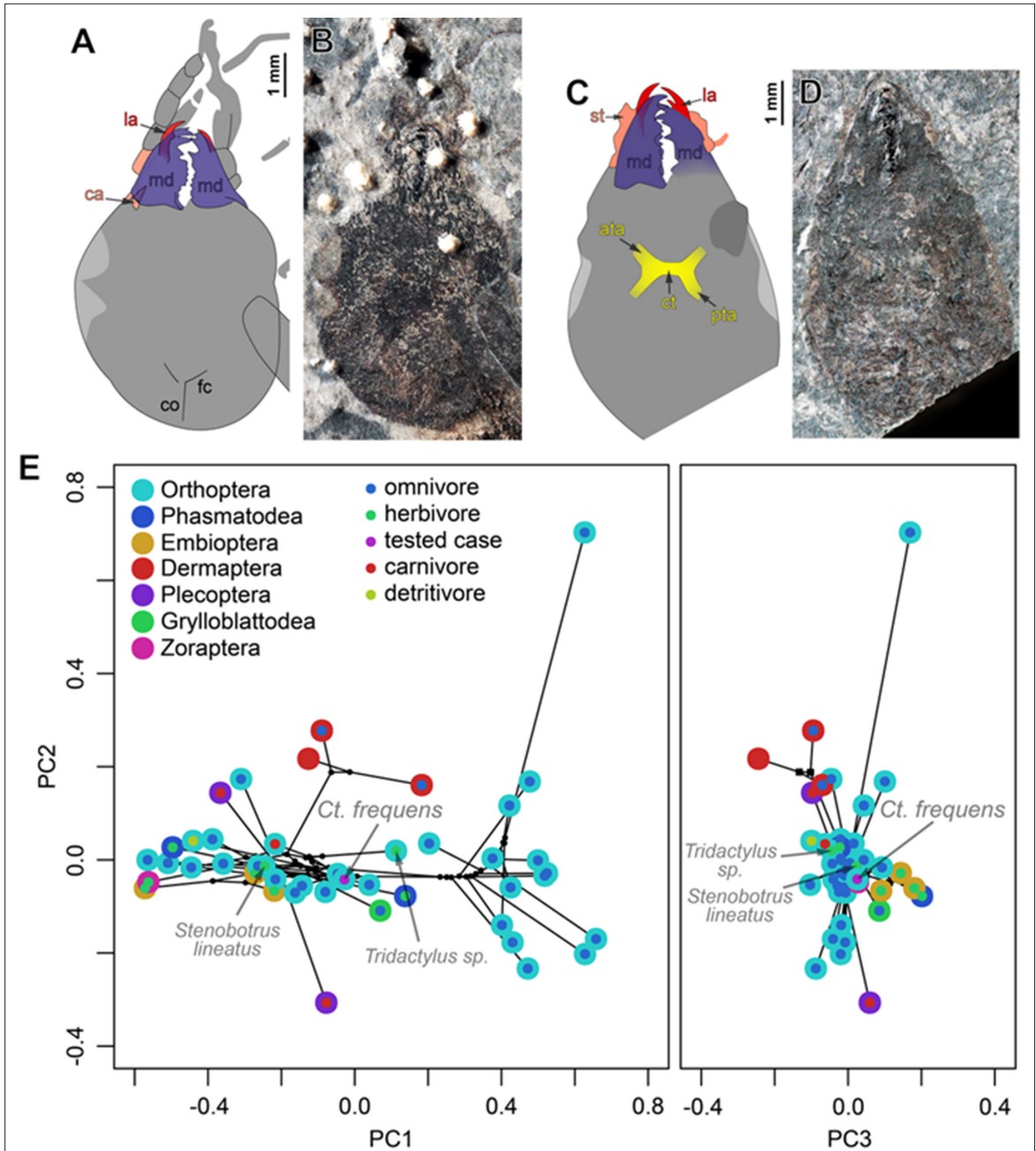

**Figure 4.** Head morphology (A–D) in *Ctenoptilus frequens* sp. nov. and (E) mandibular mandibular mechanical advantage in *Ct. frequens* sp. nov. and a selection of polyneopteran species. (A–B) Specimen CNU-NX1-754, (A) color-coded interpretative drawing, and (B) photograph (composite) (as located on *Appendix 1—figure 7I*); (C–D) Specimen CNU-NX1-764, (C) color-coded interpretative drawing, and (D) photograph (composite). (E) Principal component analysis of the mandibular mechanical advantage. Color-coding: (A–D) red, lacina (la); salmon, cardinal and stipital sclerites (ca and st, respectively); dark blue-purple, mandible (md); yellow, tentorium, including anterior tentorial arm (ata), posterior tentorial arm (pta), and corpotentorium (ct). Other indications: co, coronal cleavage line; fc, frontal cleavage line.

crown-Orthoptera and their nearest stem relatives (*Béthoux and Nel, 2002*). Given this evidence, based on the configuration of several body parts, *Ct. frequens*, and its various Pennsylvanian relatives collectively referred to as 'lobeattid insects' are stem relatives of Orthoptera (*Figure 3C*). The absence of olis2 in Caelifera therefore is the consequence of a secondary loss.

## Evolution of ovipositor morphology

Based only on extant species, the evolution of the external ovipositor in crown-Orthoptera was ambiguous due to the organizational diversity of its substructures (*Cappe de Baillon, 1920*; *Cappe de Baillon, 1922*; *Kluge, 2016*; *Thompson, 1986*; *Walker, 1919*; Appendix 1, Supplemental Text, Section 2.2). Comparison has traditionally been made between Grylloblattodea (rock-crawlers) and Orthoptera (*Walker, 1919*) even though the two groups are not closely related (*Wipfler et al., 2019*). In both groups the ovipositor displays an elongate gs9 and a ball-and-socket locking mechanism, the so-called primary olistheter (olis1), interlocking gp9 onto gp8 (*Figure 2G*). This olis1 occurs widely among insects (*Klass, 2008*). Orthoptera possess a variety of additional olistheters, including one interlocking gs9 onto gp9 (royal blue in *Figures 2 and 3*; olis2), commonly present in ensiferans possessing a well-developed ovipositor, as exemplified by Rhaphidophoridae (cave crickets; *Figure 2E*, igure 3A and B, and see sections labelled 'a–c' on *Figure 3C*), and Gryllacrididae (raspy and king crickets) and Anostostomatidae (king crickets) (*Figure 2* G3C). The occurrence of an olis2 is diagnostic of ensiferan ('sword-bearing') Orthoptera (*Kluge, 2016*; and see below).

Even though it is unclear how far posteriorly olis2 extends in *Ct. frequens*, the asserted phylogenetic placement of this species provides new insights on the evolution of ovipositor interlocking mechanisms in Orthoptera (*Figure 3*). The one in *Ct. frequens* is best comparable to the one of Rhaphidophoridae, the main difference concerning the rachis ('ball' as in 'ball-and-socket'), which is limited to a short protrusion in these insects, while the aulax ('socket' as in 'ball-and-socket') extends further posteriorly. In addition, gs9 extends more ventrally, concealing gp8 for some distance. Compared to Gryllacrididae the only notable difference in *Ct. frequens* is the ventral extension of gs9 in the former. In Anostostomatidae, the ventral margin of gs9 enters a socket in gp8, regarded as composing the premises of a third olistheter (olis3). The most parsimonious hypothesis is that this new structure ultimately replaces olis2 in Tettigoniidae and thereby allows a coupling of gs9 with gp8.

Grylloidea (true crickets) and *Ct. frequens* are separated by more severe morphological differences. A gp9 is not present in all Grylloidea and, if present, it occurs at the ovipositor base and is reduced compared to, for example, Rhaphidophoridae. Gs9 and gp8 are connected by an olistheter and we suggest that it might represent a variant of olis2, assuming a hypothetical case (shaded scheme in *Figure 3C*) in which olis2 interlocks gs9, gp9, and gp8 altogether. The reduction of gp9 would then mean that only olis2 connects gs9 and gp8. The alternative is a convergent acquisition of an olis3, as in Tettigoniidae.

Unlike other orthopterans displaying a well-developed external ovipositor, Caelifera use valves for digging a tunnel to accommodate their entire abdomen and, additionally, dig egg pods (*Fedorov, 1927*; *Stauffer and Whitman, 1997*; *Uvarov, 1966*). The shoving operation to move forward is accomplished by powerful, rhythmic, dorso-ventral openings and closings of two sets of valves (*Thompson, 1986*), gs9 and gp8+ gp9, the two latter ones being interlocked via olis1. Even though gp9 is often reduced, it plays an important role in the closing of the ovipositor via muscles attached to it (*Thompson, 1986*). Obviously, an olistheter interlocking gs9 and gp8 (i.e. olis2) would impede such movements. Given the ovipositor configuration and phylogenetic placement of *Ct. frequens*, it follows that the olis2 was lost in Caelifera, a likely consequence of their highly derived oviposition technique.

The evolutionary scenario resulting from our findings in *Ct. frequens* addresses a long-standing debate on the respective position of the two main lineages of Orthoptera, Ensifera and Caelifera. On the basis of early, fossil Saltatoria/Orthoptera displaying elongate ovipositors, palaeontologists already assumed that caeliferans derived from ensiferans (*Sharov, 1968*). However, the placement of the corresponding fossils remained contentious, leaving it possible that both, Ensifera and Caelifera, derived from an earlier, unspecialized assemblage (*Ander, 1939*). The discovery of an elongate ovipositor in the stem-orthopteran *Ct. frequens* provides a definitive demonstration that caeliferans derived from ensiferans. Because rock-crawlers can also be understood as possessing an elongate ovipositor, which would render the term 'Ensifera' ambiguous, it is proposed to coin a new taxon name, Neoclavifera, to encompass species bearing an olis2, that is, all extant orthopterans and their stem relatives as currently known (*Figure 3C*; Appendix 1, Section 2.1.1).

Another important input on the early evolution of orthopterans regards the abundance of lobeattids. Indeed, these insects are emerging as the main component of the Pennsylvanian insect fauna. They have been reported in high numbers from all major Pennsylvanian deposits (*Béthoux, 2005c*; *Béthoux, 2008*; *Béthoux and Nel, 2005a*; and Appendix 1, Section 2.1), such as *Miamia bronsoni*

at Mazon Creek (*Béthoux, 2008*). At Xiaheyan, they collectively account for more than half of all insect occurrences (*Trümper et al., 2020*). Besides a high abundance, lobeattids and other stem-orthopterans compose a species-rich group at Xiaheyan, where they represent about a third of all insect species currently known to occur at this locality (Appendix 1, Section 3, taxon Archaeorthoptera). Orthoptera, which represent the bulk of extant polyneopteran insect diversity, therefore must have diversified early during their evolution.

## Ovipositor shape and use

Extant Orthoptera resort to a wide diversity of substrates where to lay eggs, including ground, decaying leaves or wood, and stems or leaves of living plants (*Cappe de Baillon, 1920*; *Cappe de Baillon, 1922*; *Ingrisch and Rentz, 2009*; *Rentz, 1991*). This operation aims at ensuring a degree of moisture conditions suitable for eggs to fully develop, and providing protection, for example against predation. Ground is the preferred substrate of the majority of Orthoptera, including Caelifera (*Agarwala, 1952*; *Stauffer and Whitman, 1997*; *Uvarov, 1966*; and see above). Within this group, the epiphytic and endophytic habits of several, inner lineages represent derived conditions (*Braker, 1989*; *Ramme, 1926*). This habit translates into finely serrated ovipositor valves, including gs9.

As for 'ensiferan' Orthoptera, they generally possess a pointed and elongate ovipositor used to insert eggs in various substrates. In Grylloidea (including true crickets), females insert eggs in the ground using a needle-like ovipositor, or deposit them in subterranean chambers or burrows adults may inhabit, in which case the ovipositor is usually reduced (*Cappe de Baillon, 1922*; *Loher and Dambach, 1989*; *Otte and Alexander, 1983*). However, within Grylloidea, three groups, the Trigonidiinae (sword-tail crickets), the Aphonoidini, and the Oecanthinae (tree crickets), evolved oviposition in plants. In the former, which lay eggs in soft plant material, gs9 displays serration in its distal third, along its dorsal edge (*Kim, 2013*; *Otte and Perez-Gelabert, 2009*). In contrast, both Aphonoidini and Oecanthinae lay eggs in more robust plant material, translating into apices of gs9 provided with strongly sclerotized sets of teeth and hooks (*Loher and Dambach, 1989*). In Oecanthinae, in which oviposition functioning was studied in most detail, the alternate back and forth movements of gp8 induce apices of gs9 to alternately approximate and diverge (*Dambach and Igelmund, 1983*), and therefore act as a shoving tool.

The Rhaphidophoridae commonly lay eggs into the ground, or, alternatively, into rotten leaves or wood (*Hubbell, 1936*). In the latter case, the ovipositor is often curved. Interestingly, *Ceuthophilus* spp. use the ovipositor tip, somewhat truncated, to rake ground surface above oviposition holes (*Hubbell and Norton, 1978*), presumably to hide them. Anostostomatidae lay eggs in the ground or on walls of subterranean chambers (*Monteith and Field, 2001*; *Stringer, 2001*). These preferences also apply to both Gryllacrididae (*Hale and Rentz, 2001*; *Morton and Rentz, 1983*) and

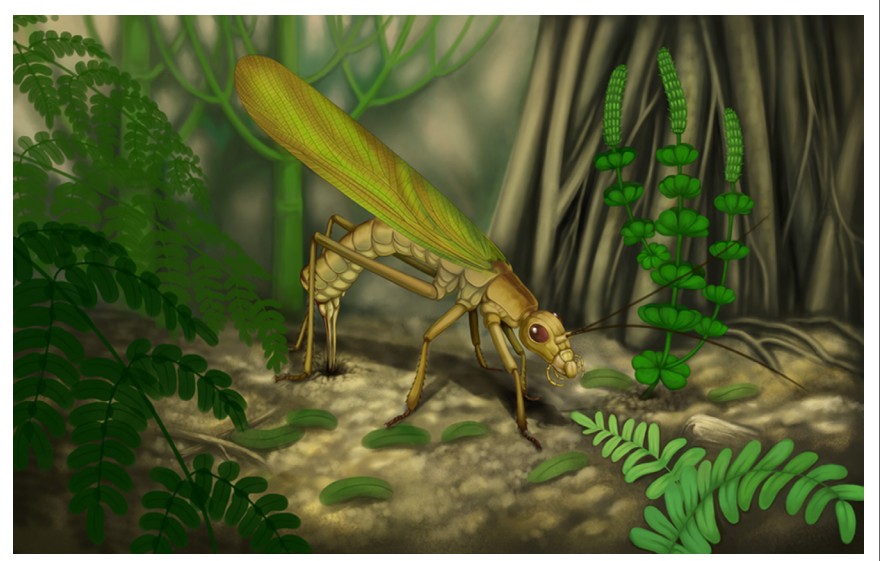

**Figure 5.** Reconstruction of a female of *Ctenoptilus frequens* sp. nov. laying eggs. Courtesy of Xiaoran Zuo.

Stenopelmatidae (*Davis, 1927*; not represented in *Figure 3C*), in which the ovipositor, if well developed, is long, narrow, and rectilinear to curved (*Cadena-Castañeda, 2019*; *Ingrisch, 2018*).

Although most Tettigoniidae (katydids) lay eggs in the ground, a variety of plant tissues, including galls, are also targeted by members of this very diverse family (*Cappe de Baillon, 1920*; *Gwynne, 2001*; *Rentz, 2010*). As above, shape and serration relate, to a large extent, to the preferred substrate. A needle-shaped ovipositor generally indicates preference for ground, a sickle-shaped one for plant tissues. Curved ovipositors indicate preference for decaying wood, and more strongly falcate ones, which are usually also laterally flattened (as opposed to sub-cylindrical), preference for either bark crevices or leaf tissues. Katydids laying eggs in hollow grass stems or leaf sheaths possess straight to slightly falcate, flattened, and unarmed ovipositors. Marked serration on the dorsal side of the ovipositor indicates preference for plant tissues.

Given the relation of ovipositor shape and substrate in extant species, *Ct. frequens*, with its needle-shaped ovipositor including ventrally oriented teeth, likely oviposited in the ground (*Figure 5*). It is therefore unlikely that Pennsylvanian stem-orthopterans were responsible for endophytic oviposition traces documented for this epoch (*Béthoux et al., 2004*; *Laaß and Hauschke, 2019*). More likely candidates for these endophytic egg laying are the extinct Rostropalaeoptera (*Béthoux et al., 2004*; *Pecharová et al., 2015a*).

### Dietary preferences

Unlike in an extant tropical forest, a limited proportion of Pennsylvanian plant foliage experienced external damage, in particular generalized feeding types such as margin and hole feeding. Although such damages were reported from multiple localities, they are so rare that their occurrence was considered worth being reported (*Correia et al., 2020*; *Iannuzzi and Labandeira, 2008*; *Laaß and Hauschke, 2019*; *Scott and Taylor, 1983*). Quantitative data from Pennsylvanian localities indicate that generalized external damages were indeed rare, and concentrated on pteridosperms ('seed ferns'; *Donovan and Lucas, 2021*; *Xu et al., 2018*). Such damages have been traditionally assigned to Orthoptera and their purported stem relatives (*Labandeira, 1998*). Indeed, investigation of mouthparts morphology in a subset of these insects suggested that, at least for the representatives belonging to the Panorthoptera/Saltatoria (*Figure 3C*), these insects were herbivores (*Labandeira, 2019*). However, there is an inconsistency between the paucity of damage on Pennsylvanian plant foliage on the one hand, and the abundance of lobeattid insects on the other. If these insects were all external foliage feeders, evidence of such damage would be more prevalent.

Given the reconstruction of the mandibular gnathal edge and its position in PC space in relation to other Orthoptera and Polyneoptera (*Figure 4E*; Appendix 1, Section 2.3), *Ct. frequens* was likely an omnivore species – not a solely herbivorous or carnivorous one. The new species is the second most common insect species at Xiaheyan, where it occurs in all fossiliferous layers at a rate of ca. 10 %. This implies that a significant portion of Pennsylvanian neopteran insects were opportunistic, omnivorous species, which reconciles the paucity of foliage damage with the abundance of stem-Orthoptera.

## Materials and methods
### Fossil material

The studied specimens are housed at the Key Laboratory of Insect Evolution and Environmental Changes, College of Life Sciences, Capital Normal University, Beijing, China (CNU). All specimens were collected from the locality near Xiaheyan village, where insect carcasses deposited in an inter-deltaic bay (*Trümper et al., 2020*).

The adopted morphological terminology is detailed in Appendix 1, Section 1.1. Documentation methodology is detailed in Appendix 1, Section 1.2.1. General habitus was investigated based on a selection of 23 specimens (including the holotype; Appendix 1, Section 2.1.2). Ovipositor morphology was investigated based on four specimens (Appendix 1, Section 1.2.2). Head and mouthparts morphology was investigated based on six specimens (Appendix, 1 Section 1.2.3).

To ensure an exhaustive documentation of ovipositor, head and mouthparts morphology, we also computed RTI files for details of several specimens. RTI files are interactive photographs in the sense that light orientation can be modified at will. The approach, originally developed in the field of archaeology (see *Earl et al., 2010* and references therein), has also been applied to a variety of sub-planar

fossil items (*Béthoux et al., 2016*; *Hammer et al., 2002*; *Jäger et al., 2018*; *Klug et al., 2019*; among others).

We computed RTI files based on sets of photographs obtained using a custom-made light dome as described elsewhere (*Béthoux et al., 2016*), driving a Canon EOS 5D Mark III digital camera coupled to a Canon MP-E 65 mm macro lens. Sets of photographs were optimized for focus using Adobe Photoshop CC 2015.5. RTI computing was then performed using the RTIbuilder software (Cultural Heritage Imaging, San Francisco, CA) using the HSH fitter (a black reflecting hemisphere placed next to the area of interested provided reference). Several snapshots were extracted using the RTIviewer software (Cultural Heritage Imaging, San Francisco, CA), including those in 'normals visualization' mode, which provides a color-coded image according to the direction of the normal at each pixel (i.e. the direction of the vector perpendicular to the tangent at each pixel; see *Figure 2C and F*). This allows to quantify subtle height differences in fossilized structures.

## Comparative analyses

The phylogeny adopted for comparative analyses is based on the most comprehensive account to date (*Song et al., 2020*), which is largely consistent with previous analyses (*Song et al., 2015*; *Zhou et al., 2017*), except for the position of the Rhaphidophoridae, either regarded as sister group of the remaining Tettigoniidea or of a subset of it. The same applies to the Schizodactylidae (splay-footed crickets), which lack a developed ovipositor.

Fossil ovipositor morphology was compared to original material of extant species and to literature data (Appendix 1, Sections 1.3.1, 2.2). Multiple interpretations of the fossil ovipositor morphology were considered. Among these, the favoured interpretation is the only one consistent with observations made on all specimens.

The MA of the mandibles, that is, the inlever to outlever ratio, indicates the effectivity of force transmission from the muscles to the food item (*Appendix 1—figure 1*). Apart from force transmission, the MA can also indicate the dietary niche and feeding habits (*Blanke, 2019*; *Sakamoto, 2010*; *Westneat, 2004*). The MA was extracted from 43 extant polyneopteran species (*Appendix 1—figure 9*) including 31 orthopterans and one CFMR of the newly described fossil species (Appendix 1, Sections 1.3.2, 1.4, *Appendix 1—table 1*). The CFMR was derived from a Procrustes superimposition (R package 'geomorph' v.3.0.5; *Adams et al., 2013*) of four fossil specimens which showed low levels of overall distortion and a mandible orientation suitable for extraction of individual MAs (*Appendix 1—figure 9*). For comparison of species and inference of the dietary niche, a PCA and, due to the detection of significant phylogenetic signal, a pPCA (R package 'phytools' v.0.6–44; *Revell, 2012*) were performed (for results of the pPCA, see *Appendix 1—figure 9*, *Appendix 1—table 2*).

## Acknowledgements

We are grateful to the numerous students who collected fossil insects at Xiaheyan; to B Kondratieff, S Schoville, and J Lapeyrie for providing material for our comparative analysis of ovipositor morphology, and to V Rommevaux for mounting and preparing this material; to S Storozhenko for providing documentation; to S Randolf for photographs of NHM Wien specimens; to S Ingrish, C Hemp, and D Rentz for discussion on ovipositor morphology in relation to substrate in 'ensiferans'; to C Labandeira for discussion on the intensity of folivory during the Pennsylvanian; and to D Marjanovic and M Laurin for discussion on nomenclatural procedures. Funding: This work was supported by the National Natural Science Foundation of China (Nos.31730087, 32020103006), and the European Research Council (ERC) under the European Union's Horizon 2020 research and innovation programme (grant agreement No 754290) awarded to AB.

# Additional information

## Funding

| Funder | Grant reference number | Author |
|---|---|---|
| European Research Council | 754290 | Alexander Blanke |
| National Natural Science Foundation of China | 31730087 | Dong Ren |
| National Natural Science Foundation of China | 32020103006 | Dong Ren Olivier Béthoux |

The funders had no role in study design, data collection and interpretation, or the decision to submit the work for publication.

## Author contributions

Lu Chen, Conceptualization, Formal analysis, Investigation, Writing – review and editing; Jun-Jie Gu, Qiang Yang, Conceptualization, Writing – review and editing; Dong Ren, Conceptualization, Funding acquisition, Project administration, Supervision, Writing – review and editing; Alexander Blanke, Conceptualization, Formal analysis, Investigation, Methodology, Project administration, Resources, Software, Supervision, Validation, Visualization, Writing – original draft, Writing – review and editing; Olivier Béthoux, Conceptualization, Data curation, Formal analysis, Investigation, Methodology, Project administration, Resources, Supervision, Validation, Visualization, Writing – original draft, Writing – review and editing

## Author ORCIDs

Alexander Blanke ⓘ http://orcid.org/0000-0003-4385-6039
Olivier Béthoux ⓘ http://orcid.org/0000-0002-3178-8967

## Decision letter and Author response

Decision letter https://doi.org/10.7554/eLife.71006.sa1
Author response https://doi.org/10.7554/eLife.71006.sa2

# Additional files

## Supplementary files

• Transparent reporting form

## Data availability

Data generated or analysed during this study are included in the manuscript and supporting files. Additional supplemental data (RTI files) are available for this paper at https://datadryad.org/stash/share/dmV-cfJHy2D475lLETIdQOzZ6HpxDWlnRk6xsw2yxXc.

The following previously published datasets were used:

| Author(s) | Year | Dataset title | Dataset URL | Database and Identifier |
|---|---|---|---|---|
| Lu C, Blanke A, Gu J, Yang Q, Ren D, Béthoux O | 2021 | Ovipositor and mouthparts in a fossil insect support a novel ecological role for early orthopterans in 300 million years old forests | https://datadryad.org/stash/landing/show?id=doi%3A10.5061%2Fdryad.mgqnk98wn | Dryad Digital Repository, 10.5061/dryad.mgqnk98wn |

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

## Appendix 1

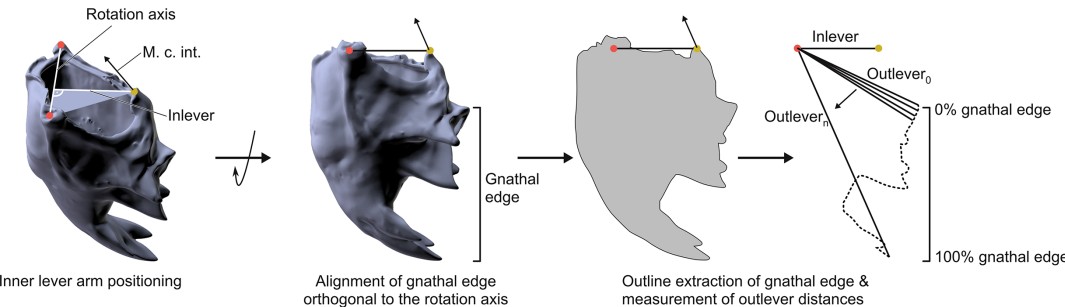

Appendix 1—figure 1. Workflow for the extraction of the mandibular mechanical advantage based on 3D models.

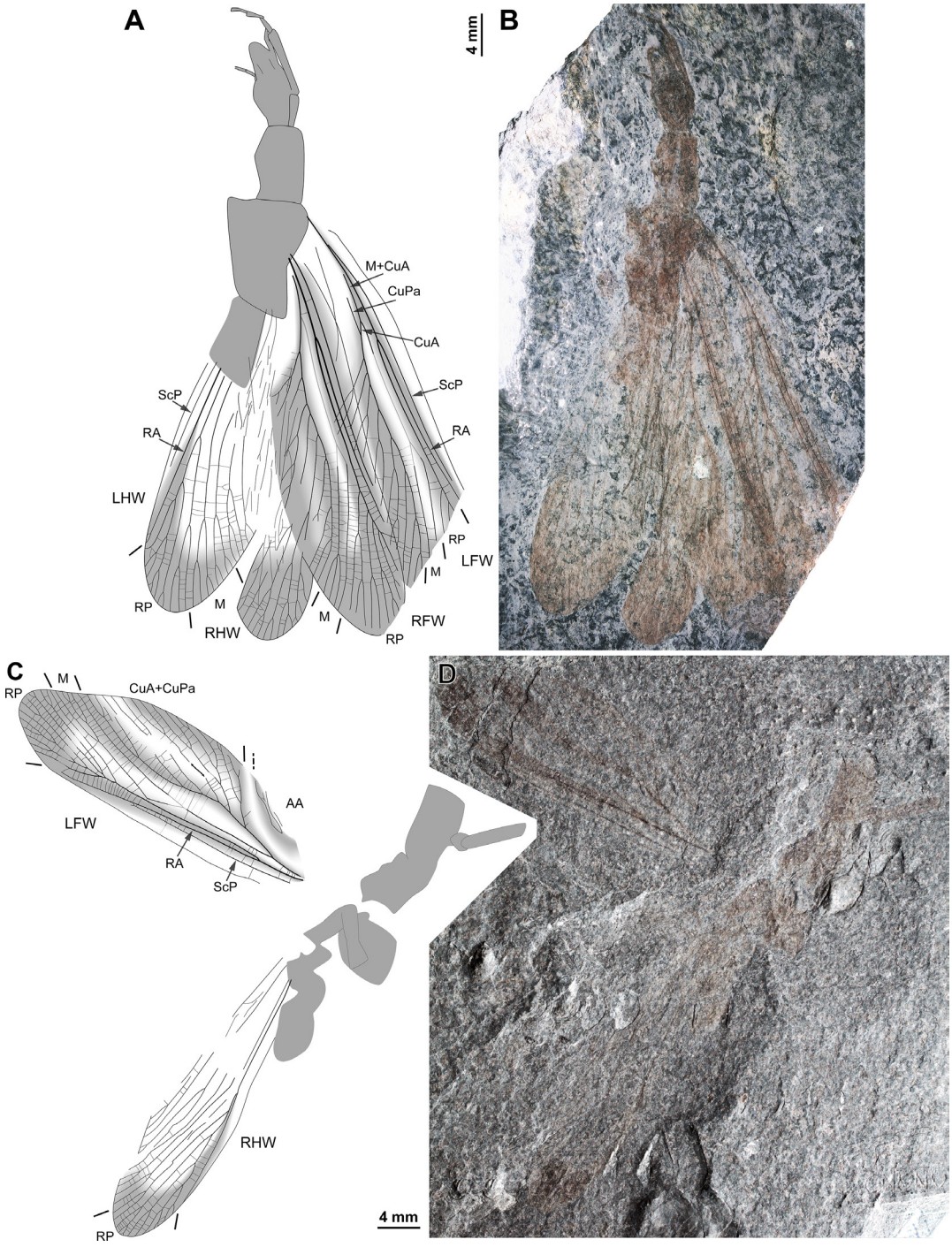

**Appendix 1—figure 2.** *Ctenoptilus frequens* sp. nov., specimens composed of fore- and hind wings in connection with body remains. (**A–B**) Specimen CNU-NX1-752; habitus, left forewing as positive imprint and right forewing and hind wings as negative imprints, (**A**) drawing and (**B**) photograph (composite). (**C–D**) Specimen CNU-NX1-738; habitus, right hind wing as positive imprints and left forewing as negative imprints, (**C**) drawing and (**D**) photograph (composite; slightly shifted vertically with respect to drawing).

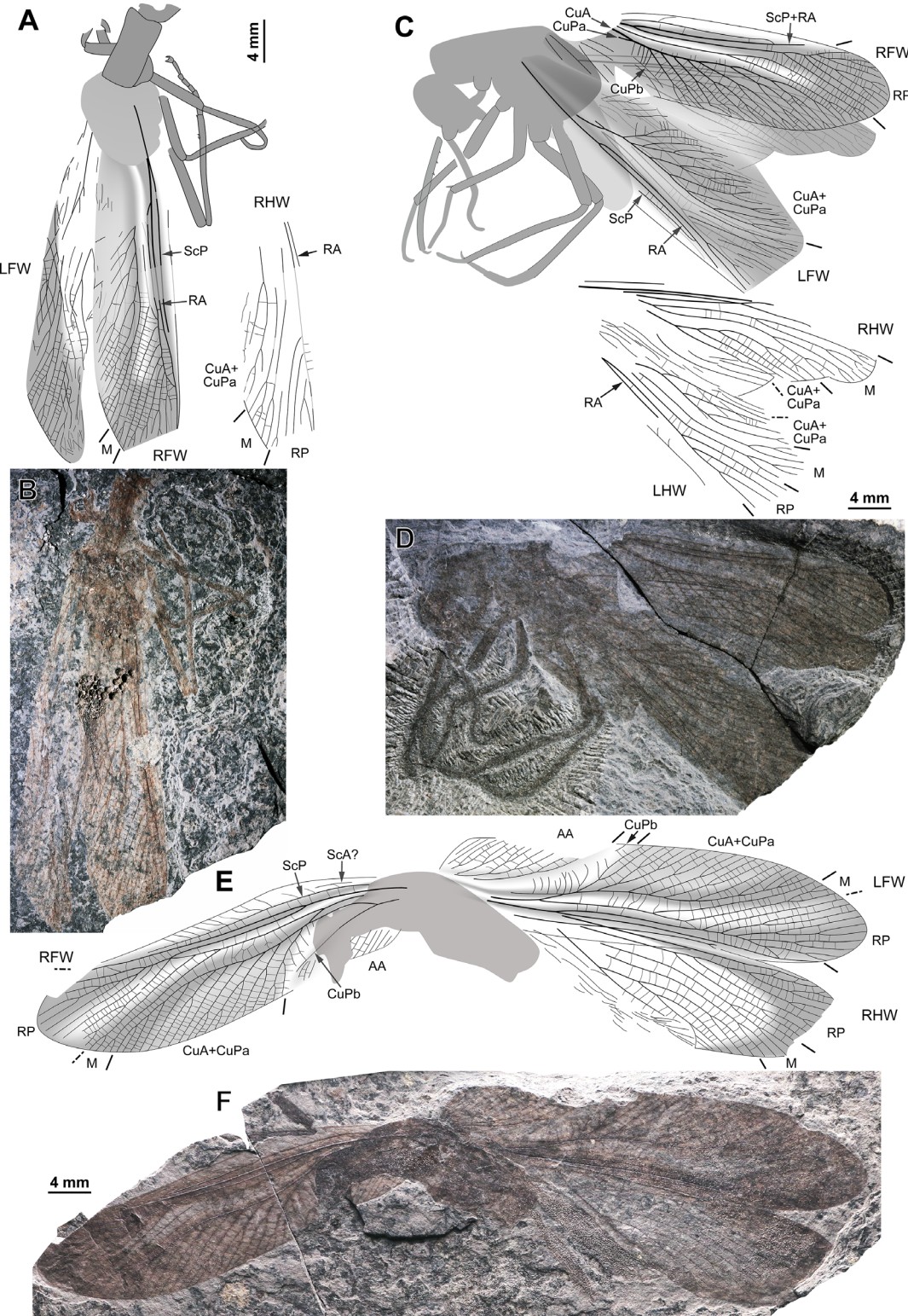

**Appendix 1—figure 3.** *Ctenoptilus frequens* sp. nov., specimens composed of fore- and hind wings in connection with body remains. (**A–B**) Specimen CNU-NX1-759; habitus, left hind wing as positive imprint and right wings as negative imprints, (**A**) drawing (for clarity, drawing of right hind wing venation duplicated and relocated, original location in light grey on complete drawing) and (**B**) photograph (composite). (**C–D**) Specimen CNU-NX1-750; habitus, all wings as negative imprints, (**C**) drawing (for clarity, drawing of hind wings venation duplicated and relocated, original location

in light grey on complete drawing) and (**D**) photograph (composite). (**E–F**) Specimen CNU-NX1-731; habitus, left forewing as positive imprint and right forewing and right hind wing as negative imprints, (**E**) drawing and (**F**) photograph (composite).

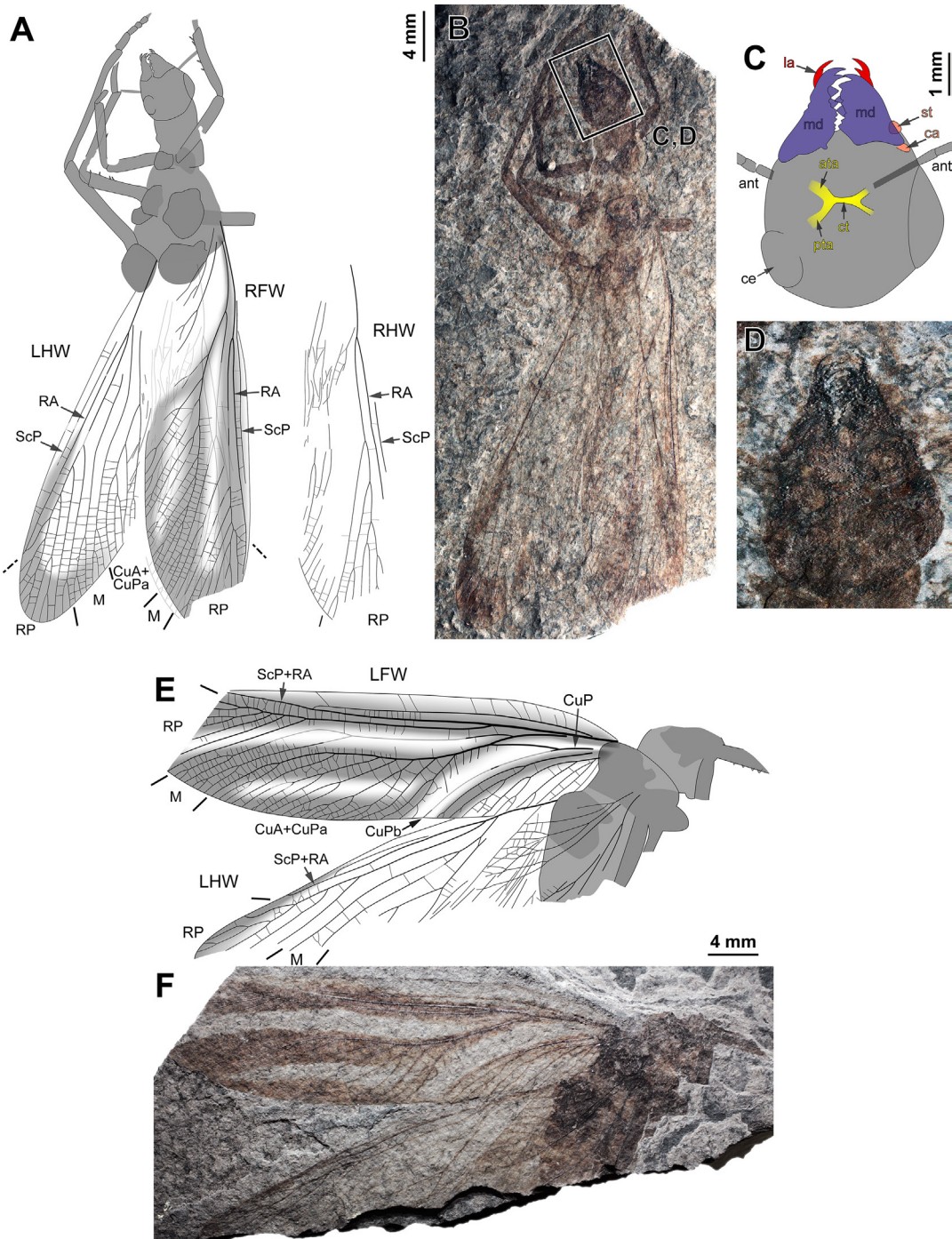

**Appendix 1—figure 4.** *Ctenoptilus frequens* sp. nov., specimens composed of fore- and hind wings in connection with body remains. (**A–D**) Specimen CNU-NX1-747; (**A–B**) habitus, all wings as negative imprints, (**A**) drawing (for clarity, drawing of right hind wing venation duplicated and relocated, original location in light grey on complete drawing) and (**B**) photograph (composite); and (**C–D**) details of head (location as indicated in **B**), polarity unclear, (**C**) color-coded interpretative drawing, and (**D**) *Appendix 1—figure 4 continued on next page*

*Appendix 1—figure 4 continued*
photograph (composite). Color-coding: red, lacina (**la**); salmon, cardinal and stipital sclerites (**ca** and **st**, respectively); dark blue-purple, mandible (**md**); yellow, tentorium, including anterior tentorial arm (**ata**), posterior tentorial arm (**pta**), and corpotentorium (**ct**). Other indications: **ant**, antenna; **ce**, composite eye. (**E–F**) Specimen CNU-NX1-741; habitus, all wings as positive imprints, (**E**) drawing and (**F**) photograph (composite).

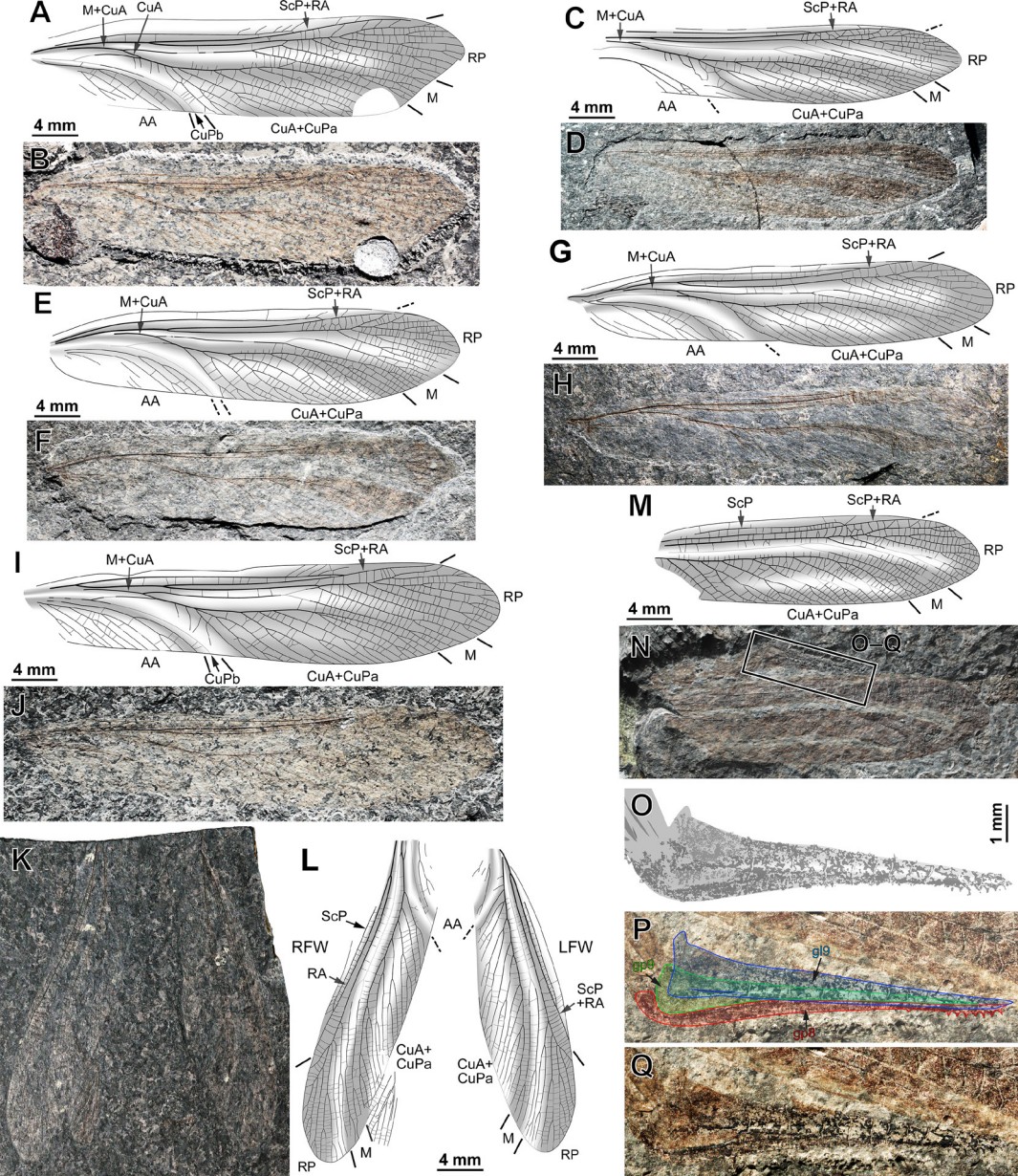

**Appendix 1—figure 5.** *Ctenoptilus frequens* sp. nov., specimens composed of forewings, isolated or by pair, and forewing and ovipositor. (**A–B**) Specimen CNU-NX1-748; right forewing, negative imprint, (**A**) drawing and (**B**) photograph (composite, flipped horizontally, light-mirrored). (**C–D**) Specimen CNU-NX1-732; right forewing, positive imprint, (**C**) drawing and (**D**) photograph (composite). (**E–F**) Specimen CNU-NX1-757; right forewing, negative imprint, (**E**) drawing and (**F**) photograph (composite, flipped horizontally, light-mirrored). (**G–H**) Specimen CNU-NX1-758; left forewing, negative imprint, (**G**) drawing and (**H**) photograph (composite). (**I–J**) Specimen CNU-NX1-744; right forewing, negative imprint, (**I**) drawing and (**J**) photograph (composite, flipped
*Appendix 1—figure 5 continued on next page*

horizontally, light-mirrored). (**K–L**) Specimen CNU-NX1-751; forewing pair, both as negative imprints, and apical fragment of a hind wing, (**K**) drawing and (**L**) photograph (composite). (**M–Q**) Specimen CNU-NX1-743; (**M–N**) habitus, right forewing, positive imprint, (**M**) drawing and (**N**) photograph (composite); and (**O–Q**) details of ovipositor (location as indicated in **N**), polarity unknown, (**O**) drawing and (**P–Q**) photographs, (**P**) with color-coded interpretative drawing and (**Q**) without (composite, flipped horizontally).

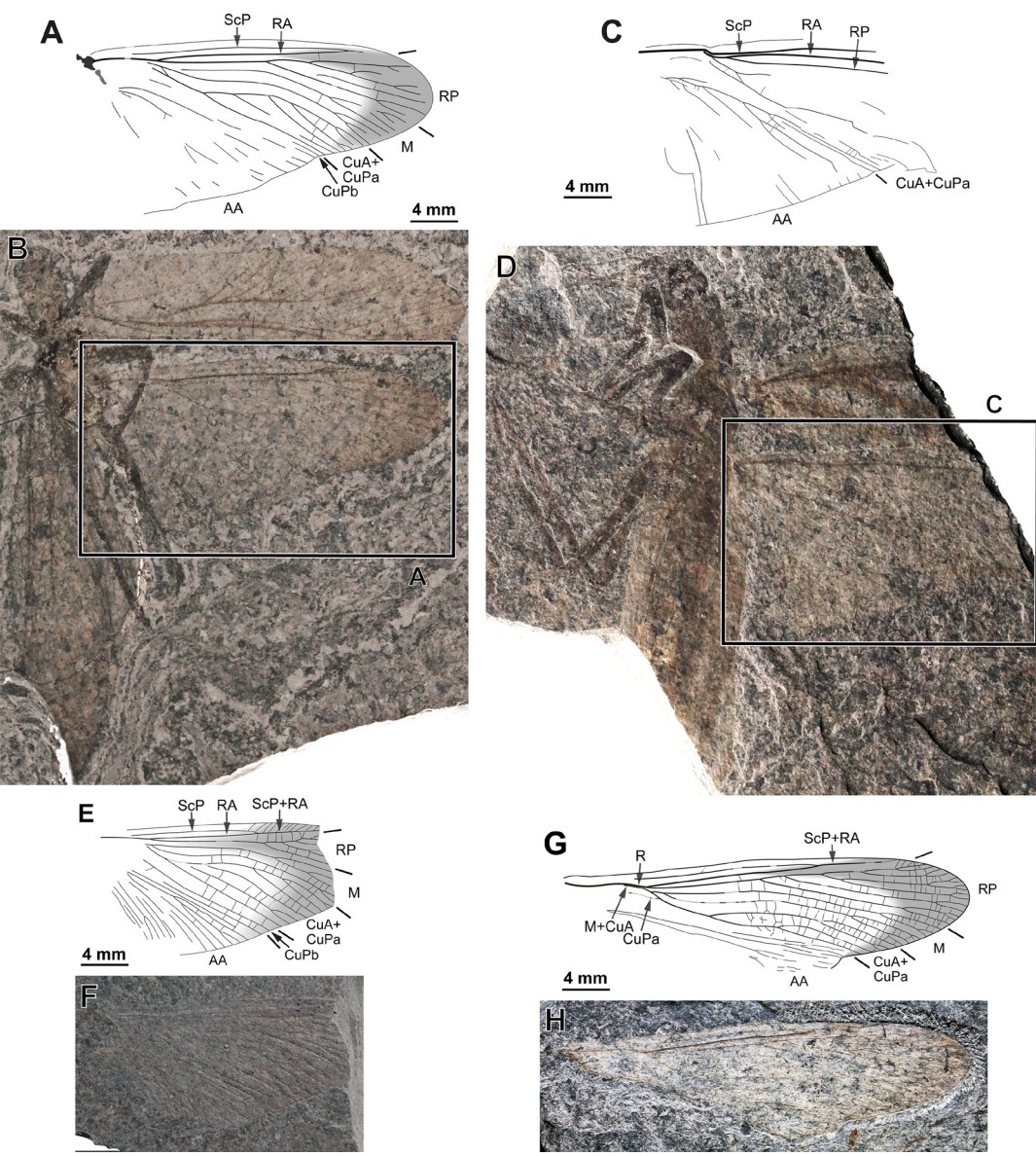

**Appendix 1—figure 6.** *Ctenoptilus frequens* sp. nov., specimens composed of well-exposed hind wings in connection with body remains or isolated. (**A–B**) Specimen CNU-NX1-198; (**A**) drawing of right hind wing (location as indicated in **B**) and (**B**) photograph of habitus (composite, flipped horizontally), left forewing as positive imprints and left hind wing and right wings as negative imprints. (**C–D**) Specimen CNU-NX1-740; (**C**) drawing of right hind wing; (**D**) photograph for habitus (composite, flipped horizontally, light-mirrored), right wings as positive imprints. (**E–F**) Specimen CNU-NX1-199; right hind wing, positive imprint, (**E**) drawing and (**F**) photograph (composite). (**G–H**) Specimen CNU-NX1-753; left hind wing, negative imprint, (**G**) drawing and (**H**) photograph (composite).

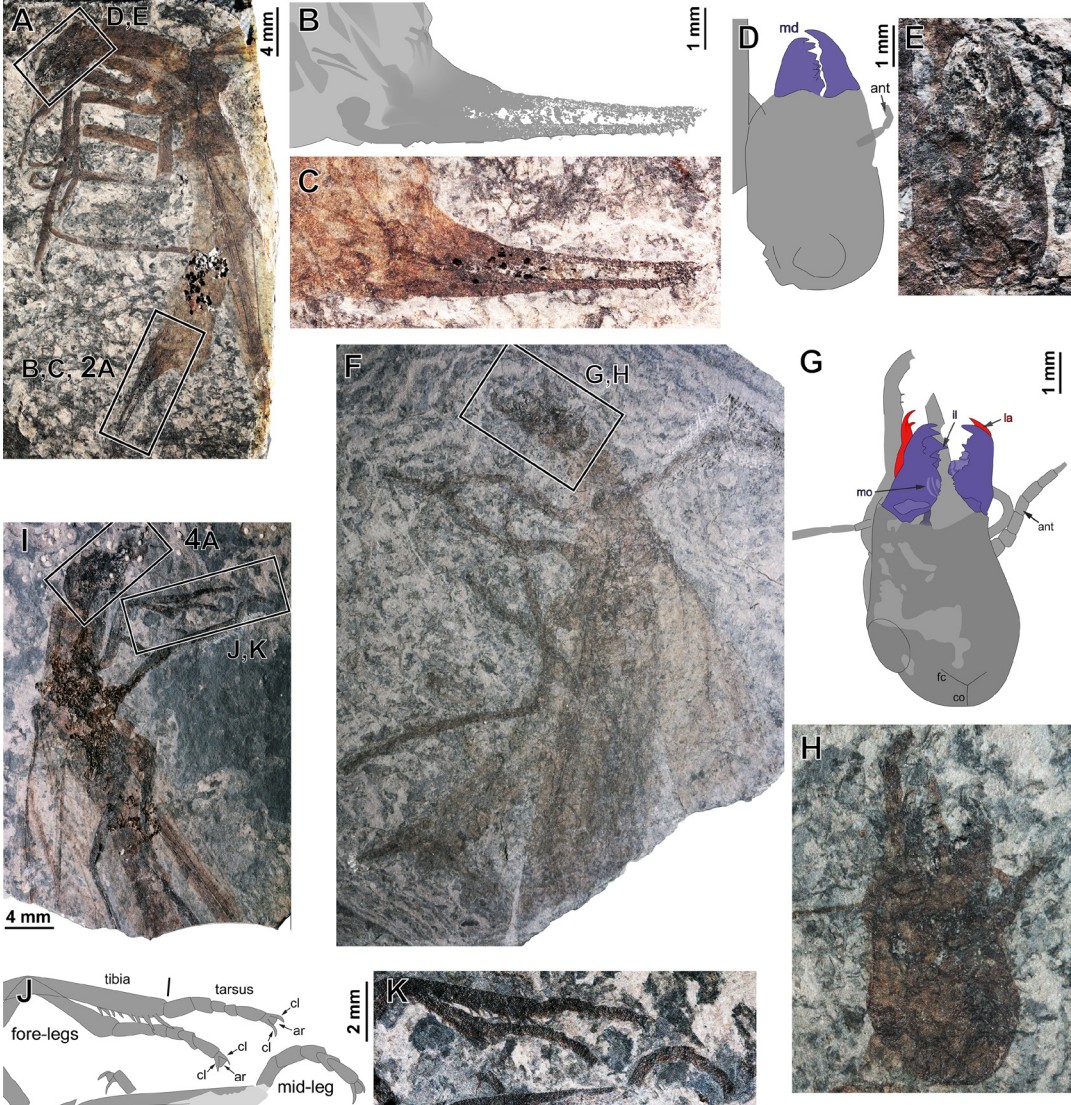

**Appendix 1—figure 7.** *Ctenoptilus frequens* sp. nov., specimens composed of body remains including well-preserved head, legs, and/or ovipositor. (**A–E**) Specimen CNU-NX1-749; (**A**) photograph of habitus (composite), left forewing as positive imprint; (**B–C**) details of ovipositor (location as indicated in A; to be compared with main document *Figure 2C*), polarity unclear, (**B**) drawing and (**C**) photograph (composite); and (**D–E**) details of head (location as indicated in A), (**D**) color-coded interpretative drawing and (**E**) photograph (composite). (**F–H**) Specimen CNU-NX1-756; (**F**) photograph of habitus (composite); and (**F–H**) details of head (location as indicated in F), imprint polarity unclear, (**G**) color-coded interpretative drawing (**F**) photograph (composite). (**I–K**) Specimen CNU-NX1-754; (**I**) photograph of habitus (composite; frame delimiting head indicating the location of main document *Figure 3A–B*), (**J–K**) details of distal portions of fore-legs and a mid-leg (location as indicated in I), (**J**) drawing and (**K**) photograph (composite).

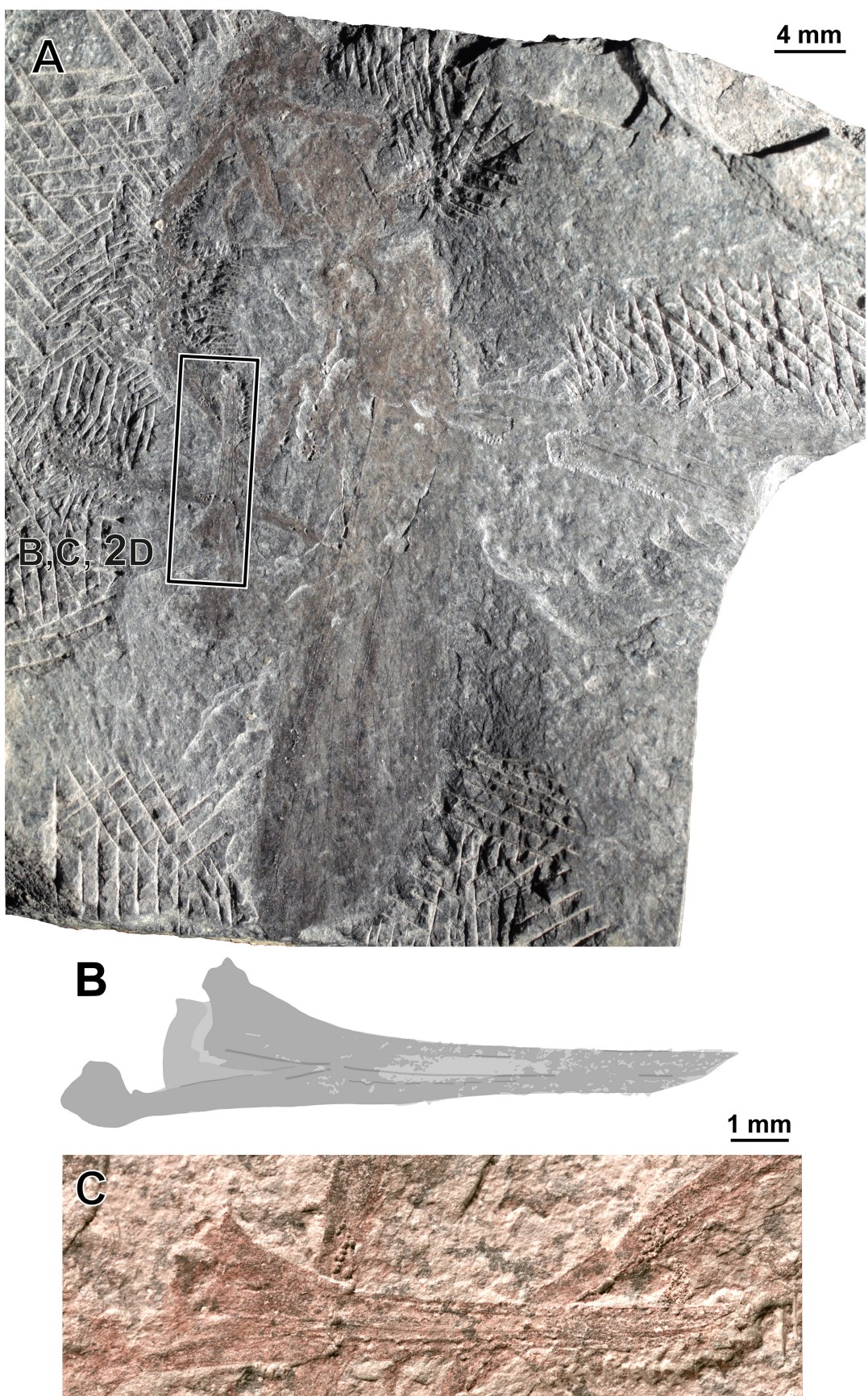

*Appendix 1—figure 8 continued on next page*

*Appendix 1—figure 8 continued*

**Appendix 1—figure 8.** *Ctenoptilus frequens* sp. nov., specimen CNU-NX1-742. (**A**) Photograph of habitus (composite), right forewing as positive imprint, flipped horizontally, and (**B–C**) details of ovipositor (location indicated in A; to be compared with main document). Photograph of habitus (composite), right forewing as positive imprint, flipped horizontally, and (**B–C**) details of ovipositor (location indicated in **A**; to be compared with main document *Figure 2D*), (**B**) drawing and (**C**) photograph (light-mirrored).

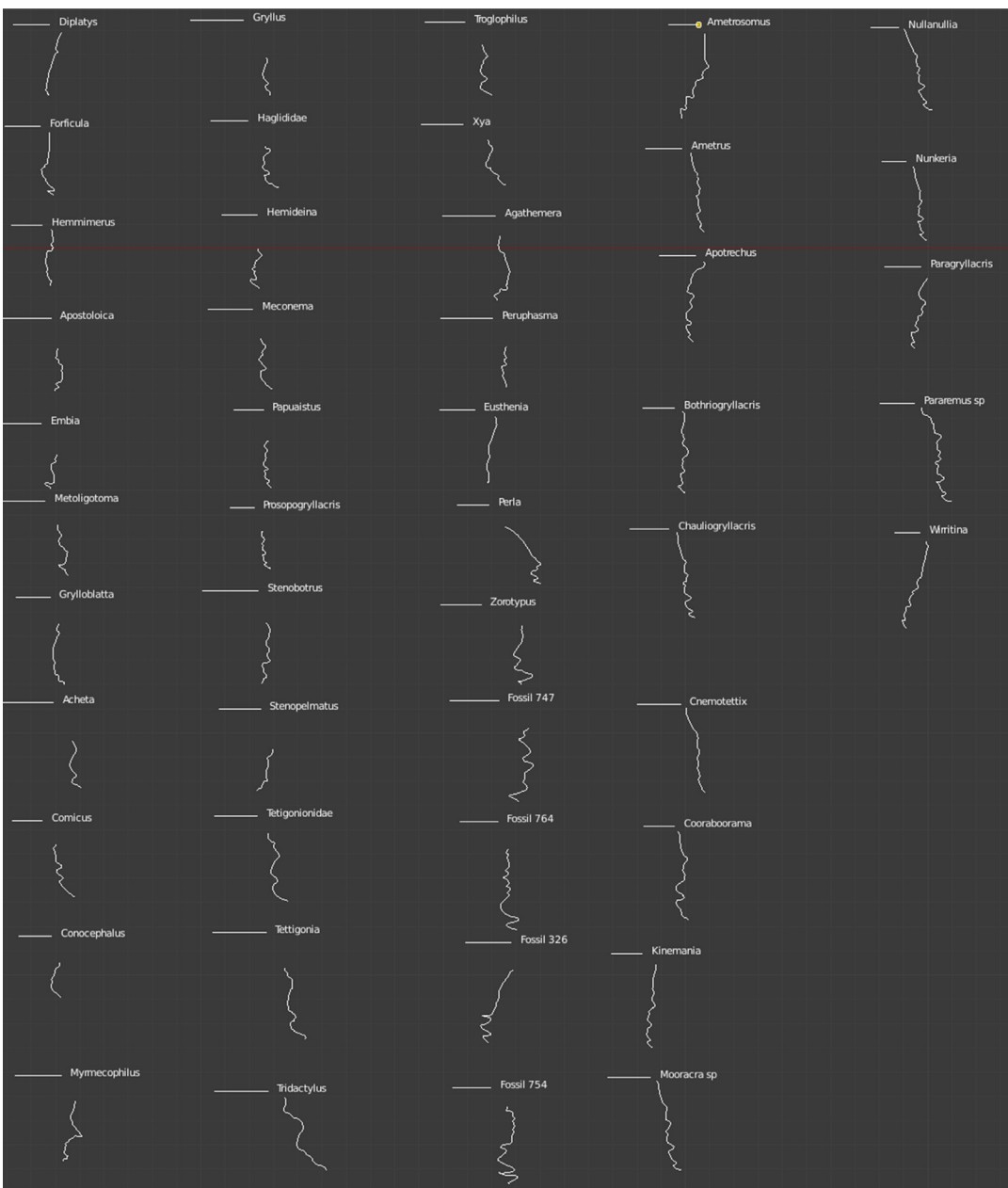

**Appendix 1—figure 9.** Outlines of the mandibular gnathal edges for all studied taxa.

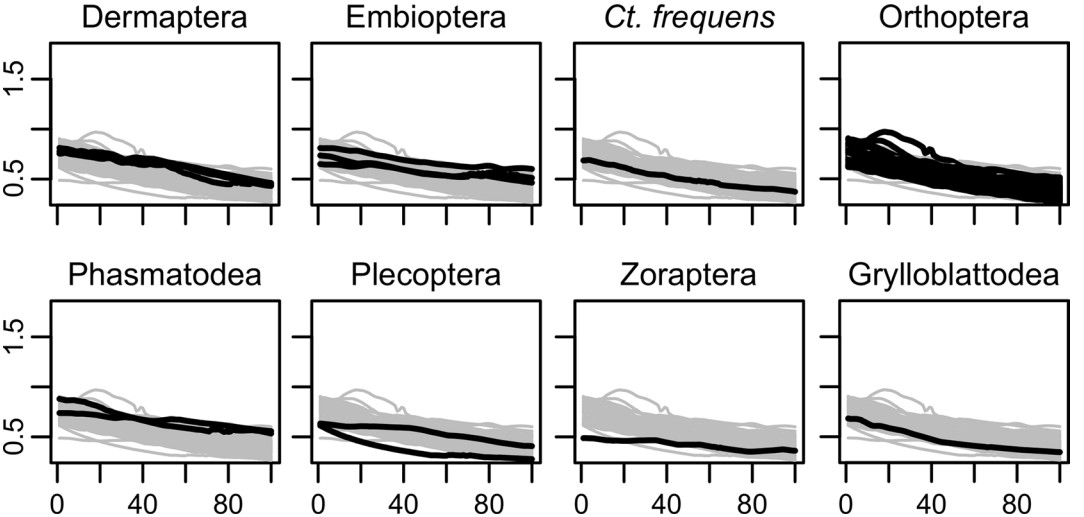

**Appendix 1—figure 10.** Progression of mechanical advantage curves for the studied taxa. x-axis = % tooth row; y-axis = MA (mechanical advantage).

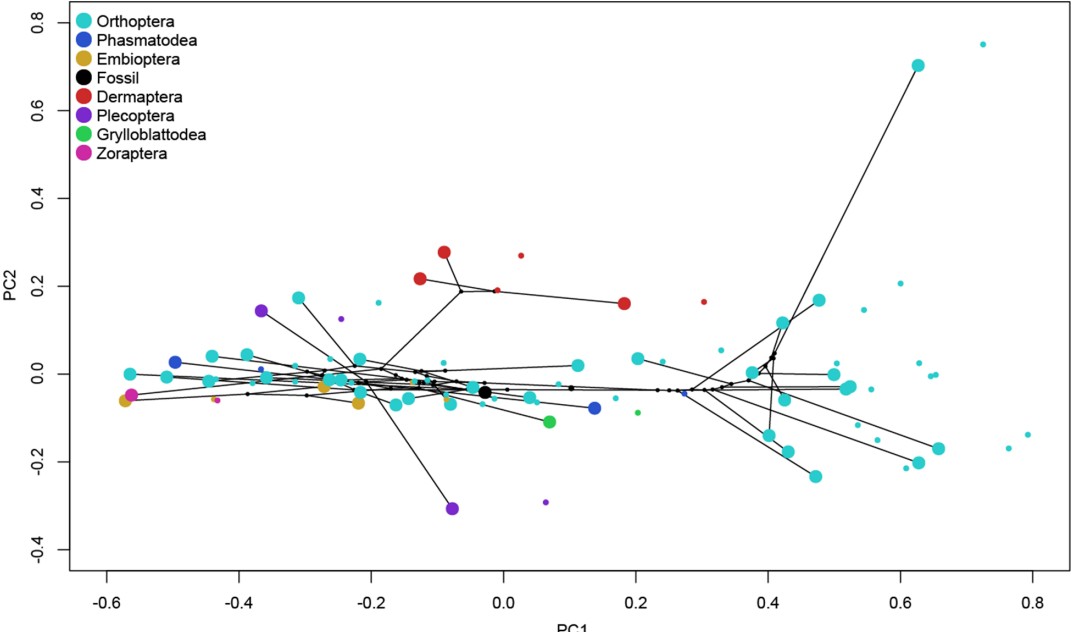

**Appendix 1—figure 11.** Results of the principal component (PC) analysis of the mandibular mechanical advantage for the first two PCs together with results for the first two PCs after phylogenetic signal correction. Large dots, distribution of species in PC space uncorrected for phylogenetic signal; small dots, distribution of species in PC space corrected for phylogenetic signal. Although phylogenetic signal was significant, differences do not affect the relative position of the sampled species to each other in PC space.

**Appendix 1—figure 12.** Correspondence between terminologies applied to polyneopteran insect ovipositors.

**Appendix 1—table 1.** Food preference of polyneopteran species included in the mechanical advantage (MA) principal component analyses.

| Order | Species | Food preference |
| --- | --- | --- |
| Dermaptera | *Diplatys flavicollis* | Omnivore |
| Dermaptera | *Forficula auricularia* | Omnivore |
| Dermaptera | *Hemimerus* sp. | Carnivore |
| Embioptera | *Aposthonia japonica* | Herbivore |
| Embioptera | *Embia ramburi* | Herbivore |
| Embioptera | *Metoligotoma* sp. | Herbivore |
| Grylloblattodea | *Grylloblatta bifratrilecta* | Omnivore |
| Orthoptera | *Acheta domesticus* | Omnivore |
| Orthoptera | *Comicus calcaris* | Omnivore |
| Orthoptera | *Conocephalus* sp. | Omnivore |
| Orthoptera | *Myrmecophilus* sp. | Omnivore |
| Orthoptera | *Gryllus bimaculatus* | Omnivore |
| Orthoptera | *Cyphoderris* sp. | Omnivore |
| Orthoptera | *Hemideina crassidens* | Omnivore |
| Orthoptera | *Meconema meridionale* | Carnivore |
| Orthoptera | *Papuaistus* sp. | Omnivore |
| Orthoptera | *Prosopogryllacris* sp. | Omnivore |
| Orthoptera | *Stenobothrus lineatus* | Herbivore |
| Orthoptera | *Stenopelmatus* sp. | Omnivore |
| Orthoptera | *Pholidoptera griseoaptera* | Omnivore |
| Orthoptera | *Tettigonia viridissima* | Omnivore |
| Orthoptera | *Tridactylus* sp. | Herbivore |
| Orthoptera | *Troglophilus neglectus* | Omnivore |
| Orthoptera | *Xya variegata* | Fetritivore |
| Phasmatodea | *Agathemera* sp. | Herbivore |
| Phasmatodea | *Peruphasma schultei* | Herbivore |
| Plecoptera | *Eusthenia lacustris* | Carnivore |
| Plecoptera | *Perla marginata* | Carnivore |
| Zoraptera | *Zorotypus caudelli* | Herbivore |

*Appendix 1—table 1 Continued on next page*

*Appendix 1—table 1 Continued*

| Order | Species | Food preference |
|---|---|---|
| Orthoptera | *Ctenoptilus frequens CFMR* | Tested |
| Orthoptera | *Ametrosomus* sp. | Omnivore |
| Orthoptera | *Ametrus tibialis* | Omnivore |
| Orthoptera | *Apotrechus illawarra* | Omnivore |
| Orthoptera | *Bothriogryllacris brevicauda* | Omnivore |
| Orthoptera | *Chauliogryllacris grahami* | Omnivore |
| Orthoptera | *Cnemotettix bifascicatus* | Omnivore |
| Orthoptera | *Cooraboorama canberrae* | Omnivore |
| Orthoptera | *Kinemania ambulans* | Omnivore |
| Orthoptera | *Mooracra* sp. | Omnivore |
| Orthoptera | *Nullanullia maitlia* | Omnivore |
| Orthoptera | *Nunkeria brochis* | Omnivore |
| Orthoptera | *Paragryllacris combusta* | Omnivore |
| Orthoptera | *Pararemus* sp. | Omnivore |
| Orthoptera | *Wirritina brevipes* | Omnivore |

**Appendix 1—table 2.** Importance and factor loadings of the principal component analyses of the polynomial regressions of the mechanical advantages (MAs).

| | **Principal component analysis** | | | | | |
|---|---|---|---|---|---|---|
| | PC1 | PC2 | PC3 | PC4 | PC5 | PC6 |
| Standard deviation | 0.380 | 0.158 | 0.083 | 0.061 | 0.039 | 0.025 |
| Proportion of variance | 0.794 | 0.136 | 0.038 | 0.021 | 0.008 | 0.003 |
| Cumulative proportion | 0.794 | 0.930 | 0.968 | 0.988 | 0.997 | 1.000 |
| Factor loadings: | | | | | | |
| | PC1 | PC2 | PC3 | PC4 | PC5 | PC6 |
| Intercept | −0.013 | 0.288 | 0.150 | 0.907 | 0.258 | 0.074 |
| Regression coefficient 1 | −0.948 | −0.250 | 0.173 | 0.046 | −0.048 | 0.065 |
| Regression coefficient 2 | 0.300 | −0.754 | 0.497 | 0.090 | 0.186 | 0.229 |
| Regression coefficient 3 | −−.037 | 0.509 | 0.528 | −0.372 | 0.290 | 0.489 |
| Regression coefficient 4 | −0.057 | −0.164 | −0.650 | −0.015 | 0.436 | 0.597 |
| Regression coefficient 5 | 0.079 | 0.028 | −0.010 | 0.171 | −0.789 | 0.585 |
| | **Phylogenetic principal component analysis** | | | | | |
| | PC1 | PC2 | PC3 | PC4 | PC5 | PC6 |
| Standard deviation | 0.745 | 0.368 | 0.172 | 0.131 | 0.099 | 0.055 |
| Proportion of variance | 0.740 | 0.180 | 0.040 | 0.023 | 0.013 | 0.004 |
| Cumulative proportion | 0.740 | 0.921 | 0.960 | 0.983 | 0.996 | 1.000 |
| Factor loadings: | | | | | | |

*Appendix 1—table 2 Continued on next page*

*Appendix 1—table 2 Continued*

| | Principal component analysis | | | | | |
|---|---|---|---|---|---|---|
| | PC1 | PC2 | PC3 | PC4 | PC5 | PC6 |
| Intercept | 1.000 | −0.021 | −0.003 | 0.003 | 0.000 | 0.000 |
| Regression coefficient 1 | −0.088 | −0.995 | −0.048 | 0.009 | 0.005 | 0.004 |
| Regression coefficient 2 | 0.054 | −0.228 | 0.940 | −0.247 | −0.008 | −0.024 |
| Regression coefficient 3 | 0.076 | 0.026 | −0.412 | −0.905 | 0.067 | −0.006 |
| Regression coefficient 4 | −0.023 | 0.067 | 0.096 | 0.101 | 0.985 | 0.078 |
| Regression coefficient 5 | 0.077 | 0.085 | 0.180 | −0.112 | −0.241 | 0.940 |

# 1 Material and methods

## 1.1 Morphological terminology and abbreviations

### 1.1.1 Head

**Ant**, antenna; **ata**, anterior tentorial arm; **ca**, cardinal sclerite; **ce**, compound eyes; **co**, coronal cleavage line; **ct**, corpotentorium; **f**, frons; **fc**, frontal cleavage line; **ga**, galea; **la**, lacinia; **il**, incisor lobe; **mo**, molar lobe; **md**, mandible; **mp**, maxillary palpus; **pha**, pharynx; **pta**, posterior tentorial arm; **st**, stipital sclerite.

### 1.1.2 Wings

We use wing venation homologies proposed by *Béthoux and Nel, 2002* for Archaeorthoptera. Corresponding abbreviations are: ScP, posterior subcosta; R, radius; RA, anterior radius; RP, posterior radius; M, media; CuA, anterior cubitus; CuP, posterior cubitus; CuPa, anterior branch of CuP; CuPb, posterior branch of CuP; AA, anterior analis; AA1, first anterior analis; AA2, second anterior analis. On figures, RFW, LFW, RHW, and LHW refer to the left forewing, right forewing, right hind wing, and left hind wing, respectively. A 'furrow' is a line along which veins and wing membrane are desclerotized. Median and cubital furrows commonly occur in insects.

### 1.1.3 Ovipositor

Several terminologies have been used to refer to the elongate and sclerotized elements (herein collectively referred to as 'valves') composing the ovipositor in insects in general (*Appendix 1—figure 12*). We favoured *Smith, 1969* terminology because it applies widely and is consensually admitted regarding homology hypotheses. In order to ease comparison, we resorted to color-coding for selected structures, as follows: light blue, gonostylus (**gs9**); light green, gonapophysis IX (**gp9**); red, gonapophysis VIII (**gp8**); royal blue, olistheter 2 (**olis2**); light orange, olistheter 3 (**olis3**); purple, *Ander, 1956* 'lateral basivalvular sclerite' (specific to Caelifera). Additional abbreviations applying to olistheter elements are as follows: olistheter 1 (**olis1**); **al**, aulax (i.e. groove, socket in 'ball-and-socket'); **rh**, rhachis (i.e. ridge, ball in 'ball-and-socket').

## 1.2 Documentation of fossil material

### 1.2.1 General aspects

Handmade draft drawings were produced using a LEICA MZ12.5 dissecting microscope equipped with the aid of a drawing tube (Leica, Wetzlar, Germany). Photographs were taken using Canon EOS 550D or 5D Mark III digital cameras (Canon, Tokyo, Japan), coupled to a Canon 50 mm macro lens, a 100 mm macro lens, or a Canon MP-E 65 mm macro lens, all equipped with polarizing filters. Each specimen was photographed under dry condition and covered with a thin film of ethanol. When available, both imprints were photographed. These photographs were optimized using Adobe Photoshop CC 2015.5 (Adobe Systems, San Jose, CA) and assembled, together with handmade drawings, into a single, multi-layered document. Reproduced photographs referred to as 'composites' are a combination of photographs of a dry specimen and the same under ethanol.

In addition to traditional photographs, we computed RTI files for details of several specimens (see main document). The corpus of data was used to produce illustrations using Adobe Illustrator CS6 (Adobe Systems, San Jose, CA). Multi-layered documents (photographs only) and RTI files are

provided in the associated Dryad dataset (*Chen et al., 2021*). Investigated specimens are listed in the Section 2.1.2.

Measurements were based on complete specimens illustrated herein and are provided in the following format: minimum/average/maximum.

### 1.2.2 Ovipositor morphology
The ovipositor morphology was investigated based on specimens CNU-NX1-326 (*Figure 1E and F*; and see related files in Dryad repository; *Chen et al., 2021*), –749 (*Figure 2A–C*, and *Appendix 1—figure 7B and C*; and see related files in Dryad repository; *Chen et al., 2021*), –742 (*Figure 2D–F*, and *Appendix 1—figure 8B and C*; and see related files in Dryad repository; *Chen et al., 2021*), and –743 (*Appendix 1—figure 5O-Q*; and see related files in Dryad repository; *Chen et al., 2021*).

### 1.2.3 Head and mouthparts morphology
The head and mouthpart morphology was investigated based on six specimens. Four of them (viz. CNU-NX1-326, –747, –754, –764) were investigated for the MA (see Section 1.4) of their mandibles. The specimens CNU-NX1-749, and –756 were excluded from the analysis because their mandibles were preserved with a slight rotation in the frontal plane; this impeding an accurate measurement of the MA (see below).

## 1.3 Documentation of extant material
### 1.3.1 Ovipositor morphology
We complemented the available literature on the morphology of female terminalia which form the ovipositor in polyneopteran lineages and in Orthoptera in particular (*Ander, 1956*; *Bradler, 2009*; *Cappe de Baillon, 1920*; *Cappe de Baillon, 1922*; *Klass et al., 2003*; *Kluge, 2016*; *Walker, 1919*; and see *Klass, 2008* and references therein) by preparation of material belonging to various extant species (see Section 2.2). External habitus was photographed under various angles. Terminalia, together with the ultimate abdominal segments, were then cut off and mounted in a polyester resin. Three to four sections were made at various levels and hand-polished. Direct observation and photographs (same equipment as above) were used to document them.

### 1.3.2 Mandible morphology
To allow for inferences about the potential feeding ecology of the fossils, the MA was studied on a phylogenetically diverse sample of extant species including several lineages of polyneopteran insects. Twenty-nine recent taxa of Polyneoptera (*Appendix 1—table 1*) were investigated using micro-computed tomography (μCT) carried out at several synchrotron facilities: Beamline BW2 and IBL P05 of the outstation of the Helmholtz Zentrum Geesthacht at the Deutsches Elektronen Synchrotron (DESY), the beamline TOMCAT at the Paul Scherrer Institute (PSI), the TOPO-TOMO beamline of the Karlsruhe Institute of Technology (KIT), and beamline BL47XU of the Super Photon Ring 8GeV (SPring-8).

## 1.4 Analysis of the mandibular MA
### 1.4.1 Introduction
The MA is a straightforward biomechanical metric which was first introduced for vertebrates (*Westneat, 1995*; *Westneat, 2004*) and was used since in studies on vertebrate and arthropod jaw mechanics (*Blanke et al., 2017*; *Cooper and Westneat, 2009*; *Cox and Baverstock, 2015*; *Dumont et al., 2014*; *Fabre et al., 2017*; *Fujiwara and Kawai, 2016*; *Habegger et al., 2011*; *Olsen and Gremillet, 2017*; *Sakamoto, 2010*; *Senawi et al., 2015*; *Weihmann et al., 2015*). The MA is defined as the inlever to outlever ratio. For dicondylic insect mandibles, the inlever is the distance between the application of the input force and the joint axis, while the outlever arm is the distance from the biting point to the joint axis (*Appendix 1—figure 1*).

The MA thus indicates the percentage of force transmitted to the food item (i.e. the effectivity of the lever system). Although more detailed investigations concerning muscular insertion angles, muscle volumes, spatial arrangements, and muscle characteristics would be needed to quantify the absolute forces applied to a given food item, the MA is a useful mechanical performance index: It allows a size-independent comparison of the relative efficiencies within the mandibular lever system and it can be readily measured in a wide array of dried museum specimens as well as

freshly collected ones. Here, we used it to assess the efficiency of the mandibular lever system of insect fossils.

Automatic segmentations of the mandibles were performed using the software ITK-snap (*Yushkevich et al., 2006*) after which STL files were imported into the software Blender (http://www.blender.org) for further processing (*Appendix 1—figure 1*). The gnathal edge was defined sensu *Richter et al., 2002* as the area from the pars molaris (proximal to the mouth opening) to the pars incisivus (distalmost tooth). Since the homology of subparts of the gnathal area is debated (*Fleck, 2011*; *Richter et al., 2002*; *Staniczek, 2000*), the gnathal outline, as seen when orienting the mandible in line with the rotation axis (*Appendix 1—figure 1*), was scaled as a percentage of tooth row length. For this, ~800 points for each specimen were wrapped against the gnathal outline in Blender and the distance between each point orthogonal to the mandibular rotation axis (=outlever) was measured. Similarly, one point was placed at the insertion point of M. craniomandibularis internus on the mandible and the distance between this point orthogonal to the rotation axis was measured (i.e. inlever). MA measurements were carried out on the segmentations of the left mandible for each specimen. All measurements and calculations were carried out in the R software environment (v. 1.1.383) using custom scripting. Separate MAs for each studied fossil were computed and combined to a CFMR using a Procrustes superimposition as implemented in geomorph v.3.0.5 in order to account for uncertainties in MA extraction due to potential distortion artefacts. From this superimposition, the mean MA shape was extracted and used together with the MAs of recent species for the further analysis steps. Polynomial functions of the 1st–20th order were fitted against all MA profiles. The AIC was used to determine the polynomial function with the best relative fit whose coefficients were then used for further analysis.

### 1.4.2 Phylogenetic signal
Phylogenetic signal was assessed using the most recent comprehensive phylogenetic estimate as a basis (*Song et al., 2020*). The phylogeny was pruned in order to contain only the taxa analysed here. The fossils were fitted into the phylogenetic estimate based on inference derived from wing venation and leg and ovipositor morphology (see main text).

Phylogenetic signal was assessed using the K statistic as implemented in geomorph v.3.0.5 (*Adams et al., 2013*) with 10,000 random permutations. This test statistic was found to be the most efficient approach to test for phylogenetic signal (*Pavoine and Ricotta, 2013*). Since significant phylogenetic signal was detected, a PCA as well as pPCA as implemented in the phytools package v.0.6–44 (*Revell, 2012*) were carried out in order to compare the analysed specimens in MA shape space.

## 2 Results
### 2.1 Systematic palaeontology
In this section the systematics at the family-group level and below conforms to the ICZN to ensure that the new species name is valid under this Code, while that above the family-group, left ungoverned by the corresponding code, conforms to the principles of cladotypic nomenclature (*Béthoux, 2007b*; *Béthoux, 2007a*), itself compliant with the PhyloCode (*Cantino and Queiroz, 2020*). Specifically, a cladotypic definition corresponds to an apomorphy-based definition using two species as internal specifiers (each being anchored to a specimen designated as type). There are minor discrepancies between cladotypic nomenclature practice on the one hand and recommendations of the PhyloCode on the other. Notably, the first author to have associated the selected defining character state and a taxon is to be acknowledged under the former procedure.

### 2.1.1 Nomenclature above family-group level
Taxon *Neoclavifera* Béthoux, new clade name (*nom*. Béthoux *n*., *dis*. *Kluge, 2016*, *typ*. Béthoux *n*.)

Registration number. 753.

### Definition
Species that evolved from the hypothetical ancestral species in which the character state 'in female ovipositor, occurrence of a locking mechanism composed of a rhachis on gonostylus IX and of an aulax on gonapophysis IX' (also called 'secondary olistheter'; as opposed to 'in female ovipositor, absence of a locking mechanism composed of a rhachis on gonostylus IX and of an aulax on gonapophysis IX'), as exhibited by *linderi Dufour, 1861* (currently assigned to *Dolichopoda*

*Bolívar, 1880*) and *artinii Griffini, 1913* (currently assigned to *Homogryllacris Liu, 2007*), was acquired.

## Abbreviated definition

∇ apo secondary olistheter (*Dolichopoda linderi* [*Dufour, 1861*] and *Homogryllacris artinii* [*Griffini, 1913*]).

## Etymology

From 'neo-', ancient Greek prefix for 'new'; 'clavis', Latin for 'key'; and '-fera', Latin suffix for 'bearing' (feminine). This is a direct reference to **olis2**, which **rh** resembles a key in cross-section.

## Reference phylogeny

The monophyly of Orthoptera, which includes all extant species sharing the defining character state of *Neoclavifera*, is beyond doubt. *Wipfler et al., 2019*, and *Song et al., 2020*, compose two recent accounts on the topic. It follows that the acquisition of the defining character state in the cladotypic species/specifiers, attested by *Cappe de Baillon, 1920*, very probably occurred once (*Kluge, 2016*). It is considered lost in Caelifera.

## Qualifying clauses

Several qualifying clauses are explicit when using a cladotypic definition, but they need to be specified for a PhyloCode usage. The name *Neoclavifera* shall be considered as invalid as that of a taxon if it occurs that (i) the defining character state was acquired by the cladotypes/specifiers convergently, (ii) the defining character state is a plesiomorphy, (iii) the cladotypes/specifiers belong to a single species, and/or (iv) the defining character state does not occur in the specifiers (unless it is secondarily lost). There is no known evidence that one of these clauses might be challenged in our case.

## Composition

All Saltatoria (including all extant Orthoptera) and 'lobeattid insects' (understood as including cnemidolestodeans) (and see Section 2.1.2, taxonomic discussion).

## Discussion

At the first glance, the name *Chopard, 1920*, appears as a suitable name. However, it is an explicit reference to the sword-like shape of the ovipositor valves in the corresponding insects, which composes a pre-occupation under cladotypic nomenclature (conversely, the taxon name Caelifera *Ander, 1936*, is an explicit reference to chisel-like shape of the valves). In other words, the name etymologically refers to a character state different from that used to define the new taxon, which makes it unavailable for the aimed purpose. The same applies to the taxon name Dolichocera *Bei-Bienko, 1964* ('long horned'; and, conversely, 'Brachycera *Bei-Bienko, 1964*' for 'short horned'), favoured by *Kluge, 2016*. Moreover, current classificatory schemes customarily regard Ensifera and Caelifera as sister groups, while our results predict that Caelifera is to be nested within Ensifera. Prolonged ambiguity on the conversion of 'Ensifera' as a defined taxon is then to be expected, not mentioning the fact that *Ensifera Lesson, 1843* is a genus name for sword-billed hummingbirds, and *Ensifera ensifera* (*Boissonneau, 1839*) its type species.

Given this situation, and the absence of a name composing a direct reference to the occurrence of **olis2**, we propose to coin a new one. Based on our literature survey, *Kluge, 2016*, is the first author to have discriminated a taxon on the basis of the defining character state only. This author stated that an **olis2** is the autapomorphy of 'Dolichocera', but the name being a direct reference to another character state (see above), it follows that a new one is needed, hence *Neoclavifera*.

The meaning of the terms 'rhachis' and 'aulax' is critical to the proposed definition. Modest elevation and groove likely made the transition from adjoined smooth surfaces to ones bearing a proper rhachis and aulax. It is therefore necessary to define rhachis and aulax, as follows: a rhachis is a projection whose base is narrower than its projected part at its widest (best assessed in cross-section), and an aulax is its counterpart.

As defined, and based on species currently known, the composition of the taxa Archaeorthoptera and Neoclavifera overlap. We hypothesize that the defining character state of Archaeorthoptera was acquired in a hypothetical ancestral species distinct from the one of

Neoclavifera, but the order of acquisition of their respective defining character states remains unknown.

### 2.1.2 Nomenclature at the family-group level and below, within the Appendix

Family Ctenoptilidae *Aristov, 2014*

 Genus *Ctenoptilus Lameere, 1917*

 *Ctenoptilus frequens Chen et al., 2020*, sp. nov.

## Etymology

Based on '*frequens*' ('frequent' in Latin), referring to the abundance of the species at Xiaheyan.

## Holotype

Holotype: CNU-NX1-326 (female individual; *Figure 1*).

## Referred material

CNU-NX1-198, −199,−731 to −759,−764 (specimens herein figured: CNU-NX1-198, −199,−731, −732,−738, −740–744, −747–754, 756–759).

## Locality and horizon

Xiaheyan Village, Zhongwei City, Yanghugou Formation (Ningxia Hui Autonomous Region, China); latest Bashkirian (latest Duckmantian) to middle Moscovian (Bolsovian), early Pennsylvanian (*Trümper et al., 2020*).

## Differential diagnosis

Compared to *Ct. elongatus* (*Brongniart, 1893*), it is most closely related species (Appendix 1, Section 2.1), smaller size (deduced from forewing length) and prothorax longer than wide (as opposed to quadrangular).

## General description

Body length (excluding antennae, including ovipositor) about 42–52 mm (based on female individuals only). **Head**: prognathous, head capsule heart-shaped in dorsal view; **md** with strongly sclerotized and prominent incisivi and a well-sclerotized molar area; **la** with a strong apical tooth and a smaller sub-apical one; **mp** well-developed, with five observed segments; tentorium composed of well-developed **ata**, **ct**, and **pta**, dorsal arms not visible; **co** located in the midline along the dorsal side of the head capsule, then branching into two diverging **fc**; **ant** long, filiform. **Thorax**: prothorax longer than wide, longer than head; boundary between mesothorax and metathorax not visible. *Wings:* ScP reaching RA distal to the two-thirds of wing length; RA with few or no anterior veinlets; RA and RP strong, parallel for a long distance; RA-RP area narrow in its basal half; at the wing base, R and M + CuA distinct; MA and MP simple for a long distance, with similar numbers of terminal branches, usually 1–3, rarely more than 4; CuA diverging from M + CuA and fusing with CuPa; CuA+ CuPa posteriorly pectinate. *Forewing*: length 31.5/**36.1**/41.2 mm, largest width 6.9/**8.3**/10.7 mm, membranous; ScP with anterior veinlets; RA-RP fork slightly distal to the point of divergence of M and CuA (from M + CuA); RP branched distally, near the second third of wing length, usually with 11–17 branches reaching apex, and occasionally 1–2 veinlets reaching RA; first split of M + CuA (into M and CuA) near the first fourth of wing length; between the origin of CuA (from M + CuA) and the first fork of RP, M very weak; first fork of M near wing mid-length; MA distinct from RP, connected to it by a short cross-vein, or occasionally fused with it for a short distance; median furrow located along M and then MP; CuA+ CuPa with most of its main branches further branched, with a total of 16–26 terminal branches; in basal part, CuA+ CuPa emitting strong posterior veinlets, vanishing before they reach the claval furrow; CuPb concave, weak and simple; AA1 with 3–4 branches; AA2 with about 10 branches; cross-veins mostly not reticulated, except along the apical and postero-apical section of the wing margin, and in the ScP/ScP+ RA RP area (where they are particularly strong); longitudinal pigmented areas located (i) along R, (ii) along CuA, and then the main stem of CuA+ CuPa, and (iii) along the posterior wing margin, distal to the endings of the first branches of CuA+ CuPa; these three areas merge distally; additional pigmented area along AA1. *Hind wing*: as in forewing, except for the following: slightly shorter than forewing; RA-RP fork opposite the point of divergence of M and CuA (from M + CuA); RP usually with 11–16 branches reaching apex; M forked at the first quarter of the wing;

M with 5–8 branches reaching posterior wing margin; CuA+ CuPa with 5–8 branches; pigmented area forming an arc covering the apex, beginning along RA and ending close to the end of CuPb; plicatum well developed, with plica prima anterior reaching the posterior wing margin opposite the end of ScP (on RA). *Legs*: Fore-leg femur 4.9–6.3 mm long, 1.0–1.3 mm wide, tibia 5.2–6.3 mm long; mid-leg femur 5.2–6.4 mm long, tibiae 5.9–7.3 mm long; hind-leg femur 7.5–11.5 mm long, tibia 9.8–12.0 mm long; spines, probably in two rows, present along the ventral side of tibia of all legs, concentrated near the apex (fore-leg, at least 12 spines; mid-leg, at least 8 spines; hind-leg, at least 15 spines); tarsus 5-segmented, second, third, and fourth segments shorter, terminal tarsal segment with paired claws and arolium (deduced from well-preserved fore-legs). **Abdomen**: abdomen about 17–23 mm long (based on female individuals only); female with a prominent sword-like ovipositor (see more detailed interpretation below and specimen descriptions).

### Specimens description

Holotype, CNU-NX1-326 (*Figure 1*) Positive and negative imprints of an almost complete female individual, viewed dorsally, very well preserved, with head, thorax, leg remains (including well-exposed fore-legs) and complete right forewing; right hind apex missing, left wings incomplete, left hind wing very incomplete, ovipositor apex concealed under right forewing. **Head**: about 6.6 mm long, 4.3 mm wide, prognathous; mandibles about 2.0 mm long, with prominent teeth at their apex; gnathal edge of right **md** clearly visible, heavily sclerotized, with the distal incisivus shorter than the subdistal ones; **mp** strong, but segments not visible; **f** large and separated to the vertex by a U-shaped line, laterally delimited to the well-developed genal area by a line; frontal and coronal sutures well-developed, located at the closest distance of the eyes to each other; eyes large, laterally protruding from the head capsule covering about half of the lateral head profile; **ant** incomplete, 6.3 mm long as preserved. **Thorax**: prothorax about 5.5 mm long, 3.7 mm wide. *Left forewing*: preserved length 22.2 mm, best width 8.1 mm; M with its two main branches preserved, CuA + CuPa with 22 terminal branches preserved. *Right forewing*: length 32.6 mm, width 9.6 mm; RP simple for 14.3 mm, with 16 branches reaching wing apex and one reaching ScP + RA; MA connected to RP by a short cross-vein, with three branches, MP with four branches; CuA + CuPa with 26 terminal branches preserved; CuPa partly preserved. *Right hind wing*: preserved length: 30.4 mm, best width 8.6 mm; plicatum creased. *Legs*: fore-leg femur about 4.9 mm long and 1.2 mm wide, tibia 6.3 mm long and 0.7 mm wide, tarsus about 5.0 mm long, tarsal segments (5), paired claws and arolium visible; mid- and hind-legs incomplete and/or not well exposed. *Legs*: spines well exposed on foreleg tibiae and distal part of a mid-leg tibia. **Abdomen**: bent (probably a consequence of decay), about 17 mm long, ovipositor viewed laterally, possibly slightly obliquely; bases of **gp8** strongly sclerotized, well visible.

### CNU-NX1-749 (Figure 2A-C, and Appendix 1—figure 7A-E)

Positive and negative imprints of an almost complete female individual, wings incomplete and overlapping, body about 45 mm long. **Head:** about 6.4 mm long, 3.5 mm wide. **Thorax:** prothorax about 5.6 mm long, 3.7 mm wide. *Legs*: fore-leg femur 4.9 mm long, 1.2 mm broad, tibia 5.8 mm long, 0.8 mm broad, tarsus about 3.8 mm long; mid-leg femur 5.9 mm long, 1.0 mm broad, tibiae 7.3 mm long, 0.8 mm broad, tarsus about 4.9 mm long; hind-leg femur 6.1 mm long, 1.1 mm broad, tibia 10.1 mm long, 0.7 mm broad; spines visible, or even well-exposed, on each exposed tibiae. **Abdomen**: about 17 mm long (excluding ovipositor); sword-like ovipositor viewed laterally, about 8.4 mm long; antero-basal apophyses of **gs9**, **gp9**, and **gp8** distinct, well delineated; near the ovipositor base, dorsal and ventral edges of **gs9** and **gp8**, and ventral edge of **gp9** well delineated; dorsal edge of **gp9** visible in the distal half of the ovipositor; **olis1** and **olis2** visible near the ovipositor base, strongly sclerotized; **olis1** located along the ventral edge of **gp9** and dorsal edge of **gp8; olis2** located close to (or along) the ventral edge of **gs9**, and laterally on **gp9**; **olis1** and **olis2** converging; ventral edge of **gp8** with teeth more prominent and densely distributed near the apex.

### CNU-NX1-742 (Figure 2C-F, Appendix 1—figure 8A-C)

Positive and negative imprints of an almost complete female individual, partly disarticulated, left forewing missing; body about 52 mm long. **Head:** detached from the rest of the body, mouthparts not discernible. **Thorax:** prothorax about 7.0 mm long, 3.6 mm width. *Wings*: a forewing and two hind wings visible, poorly preserved. *Legs*: fore-leg femur 5.7 mm long, 1.1 mm broad, tibia 6.2 mm long, 0.9 mm broad; spines well exposed on one hind-leg tibia, some visible on one fore-leg tibia. **Abdomen**: strongly bent, segments not discernible; ovipositor very well preserved, detached from the rest of the abdomen, about 9.5 mm long; antero-basal apophyses of **gs9**, **gp9**,

and **gp8** distinct, well delineated; near the ovipositor base, dorsal and ventral edges of **gs9** and **gp8**, and ventral edge of **gp9** well delineated; dorsal edge of **gp9** visible at the extreme base and in the distal half of the ovipositor; **olis1** and **olis2** visible near the ovipositor base, strongly sclerotized; **olis1** located along the ventral edge of **gp9** and dorsal edge of **gp8**; **olis2** located close to (or along) the ventral edge of **gs9**, and laterally on **gp9**; **olis1** and **olis2** converging; ventral edge of **gp8** with teeth more prominent and densely distributed near the apex.

### CNU-NX1-754 (Figure 4A and B, Appendix 1—figure 7I-K)
Positive and negative imprints of an almost complete individual, well-preserved, wings overlapping, incomplete and partly creased, end of abdomen missing. **Head:** about 6.8 mm long 4.5 mm wide; **md** with strongly sclerotized and prominent incisivi and a well-sclerotized molar area; terminal teeth of **la** visible; **ca** distinguishable; **co** located in the midline along the dorsal side of the head capsule, then branching into two diverging **fc. Thorax:** prothorax about 5.9 mm long, 4.4 mm wide. *Legs*: fore-leg femora 5.3 mm long and 1.1 mm broad, tibiae 5.2 mm long and 0.7 mm broad, tarsus about 4.0 mm long; mid-leg femur 5.4 mm long and 1.1 mm broad, tibia 5.9 mm long and 0.8 mm broad, tarsus about 4.5 mm long; fore- and mid-leg tarsi well preserved, five-segmented with paired claws and arolium; second, third, and fourth segments shorter, ventral process (projecting forward) of third and fourth segments visible; hind-leg femora 7.5 mm long; end of hind-leg tibiae missing, 7.1/5.9 mm long, 0.7 mm broad; spines well exposed on one of the forelegs tibiae. **Abdomen:** about 14 mm as preserved, segments not discernible.

### CNU-NX1-764 (Figure 4C and D)
Positive and negative imprints of an almost complete, isolated head, posterior part possibly overlapping with prothorax; mouthparts well preserved; **md** in occlusion, 2.1 mm long, 1.1 mm wide at their base, provided with strongly sclerotized and prominent incisivi and a well-sclerotized molar area; distal part of **la** visible, provided with a strong apical teeth and a smaller sub-apical one; tentorium composed of well-developed **ata**, **ct**, and **pta**, dorsal arms not visible; **ct** 1.2 mm long and 0.3 mm wide.

### CNU-NX1-752 (Appendix 1—figure 2A and B)
Positive and negative imprints of a partly incomplete individual, head and prothorax well exposed, a single fore-leg preserved, wings partly spread, right hind wing creased, most of abdomen missing. **Thorax:** prothorax about 7.0 mm long, 4.0 mm wide. *Right forewing*: preserved length 35.1 mm, width 8.8 mm. RP simple for 14.9 mm, with 12 branches preserved; M poorly preserved, MA simple, MP with three branches reaching the posterior wing margin; CuA+ CuPa incomplete, with 15 visible branches. *Left forewing*: apex missing, preserved length 32.3 mm, width 8.7 mm; M not visible in its median portion; a portion of CuPa basal to its fusion with CuA visible. *Left hind wing*: length 29.5 mm, width 10.0 mm; plicatum in resting position and creased; RP with 11 branches reaching apex; M with eight terminal branches.

### CNU-NX1-738 (Appendix 1—figure 2C and D)
Imprint of an individual with parts of prothorax and thorax preserved, a left forewing (as negative imprint) and a right hind wing (as positive imprint). *Left forewing*: length 31.3 mm, width 10.3 mm; RP simple for 15.2 mm, with 13 branches reaching wing apex; MA connected to RP by a very short cross-vein; M with a total five distal branches (MP simple); CuA+ CuPa with 15 preserved terminal branches. *Right hind wing*: partly creased, plicatum not discernible/preserved, wing base not discernible; length 29.6 mm, width 7.2 mm; RP with 13 branches reaching wing apex.

### CNU-NX1-759 (Appendix 1—figure 3A and B)
Imprint of a nearly complete individual, most of head missing, left forewing twisted, right hind wing concealed under right forewing, a mass of circular cavities probably indicates the location of abdominal remains. **Thorax:** prothorax about 6.4 mm long, 3.1 mm wide; *Right forewing*: apex missing, anal area not discernible; length 34.1 mm, best width 8.0 mm; RP simple for 13.7 mm, with 11 branches preserved; one reaching ScP + RA; MA fused with RP for 0.6 mm, with two terminal branches, MP with two branches; CuA+ CuPa not fully discernible, with 16 terminal branches preserved. *Right hind wing*: anterior wing margin and plicatum not discernible; RP with seven branches preserved; M with five branches (2,3), CuA+ CuPa incomplete, with four branches. *Legs*: left legs almost missing, right fore- and mid-leg with femur and tibia preserved; right fore-leg, femur 5.7 mm long, tibia 5.3 mm long; right mid-leg, femur 6.4 mm long, tibia 8.4 mm long;

right hind-leg, femur 8.3 mm long, tibia 11.1 mm long, tarsus about 6.6 mm long, with five tarsal segments, claws, and arolium visible; spines visible on one of the hind-leg tibiae.

### CNU-NX1-750 (Appendix 1—figure 3C and D)

Positive and negative imprints of an almost complete individual, forewings overlapping hind wings, complete set of legs, abdomen poorly preserved and incomplete. **Thorax:** prothorax about 5.9 mm long, 4.2 mm wide. *Right forewing*: preserved length 32.2 mm, best width 8.6 mm; RP simple for 14.2 mm, with 11 branches preserved; MA with two branches, MP with three branches; CuA+ CuPa with 23 terminal branches, CuPb partly visible. *Left forewing*: apex missing, posterior wing margin not discernible; RP simple for 13.49 mm, CuA+ CuPa with 18 branches preserved. *Hind wings*: apices and most of margins missing/not discernible, plicata partly unfolded, creased. *Right hind wing*: preserved length 28.3 mm; M with 10 branches reaching apex, CuA+ CuPa with six branches preserved. *Left hind wing*: basal part not discernible; preserved length 22.7 mm, width 8.7 mm; RP with four branches preserved, CuA+ CuPa with five branches preserved. *Legs*: fore-leg poorly preserved; mid-leg femur 5.4 mm long, 1.1 mm broad, tibia 6.7 mm long, 0.8 mm broad; hind-leg femur 7.9 mm long, 1.3 mm broad, tibia 12.3 mm long, 0.8 mm broad.

### CNU-NX1-731 (Appendix 1—figure 3E and F)

Positive and negative imprints of an almost complete individual, very well-preserved, legs and left hind wing missing; abdomen broken. **Head:** preserved length 6.7 mm. **Thorax:** prothorax about 4.7 mm long, 3.5 mm wide. *Left forewing*: length 39.6 mm, best width 9.1 mm; RP simple for 13.4 mm, with 10 branches reaching apex and one branch reaching with ScP+ RA; M well preserved, connect with RP by a long, oblique cross-vein; MA with three branches, MP with one branch; CuA+ CuPa with 17 branches reaching the posterior wing margin, CuPb poorly preserved. *Right forewing*: preserved length 35.9 mm, best width 9.2 mm; a vein interpretable as ScA partly preserved; RP simple for 13.1 mm, with 10 branches reaching apex and one veinlet reaching with ScP+ RA; M well preserved, connected with RP by a long, oblique cross-vein; MA with two branches, MP with two branches; CuA+ CuPa with 19 branches reaching the posterior wing margin; AA area with several branches preserved. *Right hind wing*: apex missing; RP with 10 branches directed towards apex and two veinlets reaching with ScP+ RA; MA and MP simple for a long distance, with three and two branches reaching the posterior wing margin, respectively; CuA+ CuPa with six terminal branches preserved; plicatum folded, creased.

### CNU-NX1-747 (Appendix 1—figure 4A-D)

Positive and negative imprints of an almost complete individual, left forewing and right hind-leg missing. **Head:** about 6.0 mm long, 4.4 mm wide; **md** about 1.7/2.0 mm long, 1.4 mm wide at their base; apical tip of **la** visible, **ct** 0.9 mm long 0.2 mm wide; compound eye oval; circumocular ridge well developed. **Thorax:** prothorax about 5.5 mm long, 3.3 mm wide. *Right forewing*: preserved length about 29 mm, best width 8.2 mm; RP simple for 12.3 mm, with 10 branches, two of them reaching ScP + RA; MA and MP with two branches each; CuA+ CuPa with 19 terminal branches visible. *Hind wings:* plicatum folded, with numerous anal veins, not clearly discernible. *Left hind wing*: length 30.1 mm, best width 7.8 mm; RP simple for 9.6 mm, with 11 branches reaching wing apex and a single veinlet reaching ScP + RA; MA and MP simple for a long distance, each with three branches; CuA+ CuPa posteriorly pectinate, with six terminal branches. *Right hind wing*: overlapping with right forewing, only partly discernible; RP simple for 8.3 mm, with nine branches preserved. *Legs*: right legs poorly preserved and/or incomplete; left fore-leg femur 6.3 mm long and 1.0 mm wide, tibia 5.2 mm long and 0.6 mm wide; mid-leg femur 5.0 mm long and 1.1 mm wide, tibia 6.8 mm long and 0.6 mm wide; hind-leg femur 8.6 mm and 1.0 mm wide, tibia 9.8 mm long and 0.6 mm wide; spines well exposed on both foreleg tibiae, and the preserved mid-leg and hind-leg tibiae.

### CNU-NX1-741 (Appendix 1—figure 4E and F)

Positive and negative imprints of an incomplete individual, left forewing and thorax preserved, left hind wing and right forewing incomplete. *Left forewing*: apex missing, preserved length 34.8 mm, best width 10.7 mm; RP simple for 15.6 mm, with seven branches preserved; M poorly preserved, MA and MP with three branches each; CuA+ CuPa with 23 terminal branches; AA1 with four main branches. *Right forewing*: only the basal part preserved, AA1 with four branches, AA2 with six branches preserved. *Left hind wing*: preserved length 33.1 mm; RP with six branches; MA and MP simple in the preserved part.

### CNU-NX1-748 (Appendix 1—figure 5A and B)

Well-preserved isolated right forewing, negative imprint; length 38.4 mm, best width 8.7 mm; RP simple for 14.1 mm, with 12 terminal branches reaching apex; MA with two branches, the anterior one connected to RP by a cross-vein; MP with five branches; CuA+ CuPa with 16 terminal branches reaching the posterior wing margin, and a branch fused with MP; CuPb visible, area between CuPb and AA1 narrow; AA1 with four preserved branches.

### CNU-NX1-732 (Appendix 1—figure 5C and D)

Positive imprint of nearly complete right forewing; preserved length 30.3 mm, best width 6.6 mm; RP simple for 12.8 mm, with 17 branches reaching wing apex and three branch reaching ScP+ RA; basal portion of M relatively well-preserved, first fork located opposite wing mid-length; MA simple; MP with three branches reaching posterior wing margin and a veinlet fusing with MA; CuA+ CuPa with 16 branches; AA1 with three branches.

### CNU-NX1-757 (Appendix 1—figure 5E and F)

Negative imprint of a well-preserved, isolated right forewing; length 35.3 mm, best width 8.4 mm; RP simple for 14.4 mm, with 12 branches reaching wing apex and two branches reaching ScP + RA; basal portion of M relatively well-preserved; MA and MP with four and three branches, respectively; CuA+ CuPa with 19 terminal branches reaching posterior wing margin; AA1 with four branches; AA2 poorly preserved.

### CNU-NX1-758 (Appendix 1—figure 5G-H)

Negative imprint of a well-preserved, isolated left forewing, slightly creased along the claval furrow; length 36.6 mm, best width 7.2 mm; RP simple for 13.9 mm, with 11 terminal branches reaching apex; MA and MP with two and three branches, respectively; CuA+ CuPa with 16 terminal branches reaching posterior wing margin.

### CNU-NX1-744 (Appendix 1—figure 5I and J)

Negative imprint of a well-preserved, isolated right forewing; length 41.2 mm, best width 8.7 mm; RP simple for 16.0 mm, with 12 terminal branches visible; MA connected to RP by a short cross-vein; MA and MP with two terminal branches each; CuA+ CuPa with 19 branches reaching posterior wing margin; AA1 with three preserved branches.

### CNU-NX1-751 (Appendix 1—figure 5K and L)

Negative imprint of nearly complete forewing pair and apical fragment of a hind wing. *Forewings*: basal half of M and CuPb not visible; MA and MP with two branches each. *Left forewing*: preserved length 30.4 mm, best width 6.9 mm; RP simple for 13.7 mm, with 17 branches reaching wing apex; CuA+ CuPa with 17 terminal branches, AA1 with three branches. *Right forewing*: length 27.1 mm, best width 7.4 mm; RP simple for 13.6 mm, with 14 preserved branches reaching wing apex; CuA+ CuPa with 21 terminal branches.

### CNU-NX1-743 (Appendix 1—figure 5M-Q)

Positive imprint of an incomplete right forewing and of an ovipositor. *Right forewing*: basal part missing, preserved length 26.9 mm, best width 7.8 mm; RP with 14 branches reaching wing apex and a veinlet reaching ScP + RA; M and most of MA poorly preserved; MA and MP with three branches each; CuA+ CuPa incomplete, with 17 branches reaching the posterior wing margin, and one veinlet fusing with MP. *Ovipositor*: preserved length 9.2 mm; **olis2** and dorsal margin of **gp8** strongly sclerotized; prominent teeth visible in the distal part of **gp8**.

### CNU-NX1-198 (Appendix 1—figure 6A-B)

Positive and negative imprints of an almost complete individual, head and abdomen missing, wings moderately well preserved, left wings overlapping. **Thorax:** prothorax about 6.6 mm long, 3.9 mm wide. *Right forewing*: length 32.1 mm, best width 8.8 mm; RP simple for 12.1 mm, with nine branches preserved; M poorly preserved, MA and MP with three and two branches, respectively; CuA+ CuPa incomplete, with 14 branches preserved. *Right hind wing*: length 28.7 mm, best width 14.6 mm; RP simple for 14.6 mm, with eight branches preserved; MA and MP simple for a long distance, M with seven branches reaching the posterior wing margin; fusion of CuA (emerging from M + CuA) with CuPa visible; CuA+ CuPa with seven terminal branches; CuPb partly preserved; plicatum almost fully deployed, large, probably with vannal folds; AA with nine branches preserved.

### CNU-NX1-740 (Appendix 1—figure 6C and D)

Positive and negative imprints of an incomplete individual, with forewings and right hind wing poorly preserved, abdomen not discernible. **Head:** 7.5 mm long, 4.7 mm wide. **Thorax:** prothorax about 5.5 mm long, 4.5 mm wide. *Left hind wing*: apex missing; fusion of CuA (emerging from M + CuA) with CuPa visible; plicatum well deployed, large, with several veins preserved (attributable to AA). *Legs*: fore-leg femur length 5.5 mm long and 1.2 mm wide; mid-leg femur 5.2 mm long and 1.2 mm wide; hind-leg femur 11.5 mm long and 1.2 mm wide, tibiae 12.0 mm long and 0.8 mm wide, tarsus about 6.2 mm long, paired claws and arolium preserved.

### CNU-NX1-199 (Appendix 1—figure 6E and F)

Positive and negative imprints of isolated right hind wing; wing base not discernible, apex missing; preserved length 17.4 mm, best width 10.1 mm; RP simple for 8.9 mm, with six branches preserved; M with five branches reaching the posterior wing margin; CuA+ CuPa with eight terminal branches; plicatum well deployed, with 17 branches preserved (attributable to AA).

### CNU-NX1-753 (Appendix 1—figure 6G and H)

Negative imprint of an isolated left hind wing, plicatum not discernible/preserved; length 34.2 mm, best width 10.1 mm; at the wing base, M + CuA distinct from R; RP simple for 9.5 mm, with 16 branches reaching wing apex; MA and MP simple for a long distance, with three and four branches, respectively; CuA + CuPa with five terminal branches preserved; plicatum with several visible veins (attributable to AA).

### CNU-NX1-756 (Appendix 1—figure 7F–H)

Positive and negative imprints of an almost complete female individual, wings poorly preserved and incomplete, total length (excluding **ant**) about 51 mm. **Head:** 7.1 mm long, 3.6 mm wide; **md** open; left **md** with well-discernible **il** and **mo**; left **la** with a strong apical tooth and a smaller sub-apical one; **co** located in the midline along the dorsal side of the head capsule, then branching into two diverging **fc**; **ant** long, filiform; **ce** 1.4 mm long and 0.8 mm wide. **Thorax:** prothorax about 5.8 mm long, 4.0 mm wide. **Abdomen:** length about 23 mm, segments not discernible; exposed portion of ovipositor about 5.0 mm long.

### Taxonomic discussion

The new species is closely related to a number of Pennsylvanian insects collectively referred to as 'lobeattids' and characterized by (i) an RA/RP fork located basally, (ii) an RA-RP area widening sharply distal to the end of ScP (on RA), and (iii) CuA+ CuPa with one main anterior branch posteriorly pectinate and with abundant branches (commonly, ca. 20) reaching the posterior wing margin. This assemblage includes *Eoblatta robusta* (*Brongniart, 1893*) and *Ct. elongatus* (*Brongniart, 1893*), from the Commentry locality (France); *Lobeatta schneideri Béthoux, 2005c, Anegertus cubitalis Handlirsch, 1911*, and *Nectoptilus mazonus Béthoux, 2005c*, from Mazon Creek (USA); *Nosipteron niedermoschelensis Béthoux and Poschmann, 2009*, from Niedermoschel (Germany); *Lomovatka udovichenkovi Aristov, 2015*, from Lomovatka (Ukraine); *Beloatta duquesni Nel et al., 2020*, from Avion (France); and *Sinopteron huangheense Prokop and Ren, 2007, Chenxiella liuae* Liu, *Ren et al., 2009 Longzhua loculata Gu et al., 2011* and *Protomiamia yangi Du et al., 2017* from Xiaheyan (China). The taxa *Miamia Dana, 1864*, and Cnemidolestodea are derived members of this assemblage.

Compared with known species, the new one is mostly similar to *Ct. elongatus*, *Ne. mazonus* and *Lom. udovichenkovi* owing to the elongate to very elongate shape of the forewing (presumed in the latter). A further similarity of the new species with *Ct. elongatus* and *Lom. udovichenkovi* is the occurrence of numerous posterior basal veinlets of CuA + CuPa vanishing before reaching CuPb. Strikingly, the new species and *Ct. elongatus* share a very particular forewing coloration pattern, with three longitudinally orientated, pigmented bands. We therefore propose to assign the new species to *Ctenoptilus Lameere, 1917*.

Note that *Béthoux and Nel, 2005a* identified, in one specimen of *Ct. elongatus*, a linear structure they interpreted as MP, that would indicate a basal position of the first fork of M. However, based on data on the new species and on the original descriptions of *Ne. mazonus* and *Lom. udovichenkovi*, we assume that the 'linear structure' is more likely the median furrow alone. If so, the first fork of M, in *Ct. elongatus*, might well be located closer to the middle of the forewing,

as in the new species and in *Ne. mazonus* and *Lom. udovichenkovi*. Note that this fork is located more basally in *Lom. udovichenkovi* than in the new species.

The forewing of the new species is smaller than in *Ct. elongatus*. Even though post-depositional deformation is known to have occurred at Xiaheyan and might have artificially elongated the forewing, the longest forewing of the new species is ca. 40 mm (with an average at ca. 36 mm), while *Ct. elongatus* forewings are 45–50 mm long. Note that female-biased sexual size dimorphism is known in a related species of Pennsylvanian Archaeorthoptera (*Du et al., 2017*). However, if one assumes that all known specimens of *Ct. elongatus* are males, then females of this species would be even longer. If all known specimens of *Ct. elongatus* are females, then they can be compared with the longest representatives of the new species, but the size gap remains then. Finally, it remains possible that the difference in size is due to a latitudinal gradient (with *Ct. elongatus* living in the equatorial area, the new species at higher latitude), but available data on the impact of latitude in extant insects size variation is too contentious to provide matter for a grounded comparison (*Chown and Gaston, 2010*). In summary, differences in size were considered sufficient to erect a new species.

Several specimens of the new species display a prothorax longer than wide (*Appendix 1— figure 1A, B* and *Appendix 1—figure 2*, and *Appendix 1—figure 6A, I* ), while it is more quadrangular in *Ct. elongatus* (see *Béthoux, 2009*). It should be acknowledged, however, that the proportions of the prothorax in the holotype (*Figure 1*) are similar to those of *Ct. elongatus*.

The set of specimens we investigated all share the colouration pattern typical for both *Ct. elongatus* and *Ct. frequens*. However, they display some variation in the forewing venation. The set of specimens on one hand, and, on the other, data on a few related species for which intra-specific variability was documented, demonstrate that this variability falls within the range of intra-specific variation. Lobeattid species relevant for comparison are *Lon. loculata*, *Miamia bronsoni Dana, 1864* (see *Béthoux, 2008*) and *Miamia maimai Béthoux et al., 2012b*.

Several specimens preserving a pair of sub-complete forewings (*Figure 1*, and *Appendix 1— figures 1A and B*, *2C–F*, and *4K and L* ) demonstrate that variation in the number and branching pattern of RP, M, and CuA + CuPa occur at the intra-specific level. More important variations are (i) the connection, or lack thereof, of an anterior veinlet from RP with RA, (ii) the connection, or lack thereof, of an anterior branch of MA with RP, and (iii) the connection of an anterior branch of CuA + CuPa with MP. As for (i), the set of specimens covers the complete range of variation, suggesting that it is not a character suitable to delimit species. Moreover, a similar range of variation has already been documented in *Lon. loculata* and *Miamia* spp. As for (ii), again, the set of specimens covers the complete range of variation of the character, with 'an anterior branch of MA and RP distinct' (*Appendix 1—figure 3C and D*, and  *Figure 5K and L*), 'an anterior branch of MA and RP connected by a short cross-vein' (*Figure 1*, and *Appendix 1—figure 2C and D*, and  *Figure 5A and B*), 'an anterior branch of MA and RP briefly connected' (*Appendix 1—figure 5I and J*), and 'an anterior branch of MA and RP fused for some distance' (*Appendix 1—figure 3A and B*). Again, the same range of variation has been documented in *Lon. loculata* and *M. maimai*. As for (iii), the trait is very rare (*Appendix 1—figure 4A and B*). Given the above and the variation documented in *Lon. loculata*, it is of very minor relevance. In summary, observed differences in forewing venation are not sufficient to distinguish distinct species.

We assign several isolated hind wings, specifically the specimens CNU-NX1-199 (*Appendix 1— figure 6E and F*) and CNU-NX1-753 (*Appendix 1—figure 6G and H*) to *Ct. frequens* because they share the same size and the distinctive colouration of *Ct. frequens* hind wings, as documented from the holotype (*Figure 1*) and other specimens preserving both fore- and hind wing, specifically CNU-NX1-752 (*Appendix 1—figure 2A and B*), CNU-NX1-738 (*Appendix 1—figure 2C and D*), CNU-NX1-731 (*Appendix 1—figure 3E and F*), CNU-NX1-747 (*Appendix 1—figure 4A and B*), and CNU-NX1-198 (*Appendix 1—figure 6A and B*). The specimen CNU-NX1-764 (*Figure 3C and D*) is an isolated head. Compared with other species occurring at Xiaheyan, it can be confidently assigned to *Ct. frequens* based on its size, shape, and features of the mandibles. The specimens CNU-NX1-749 (*Appendix 1—figure 6A–E*), CNU-NX1-756 (*Appendix 1—figure 6F–H*), CNU-NX1-754 (*Appendix 1—figure 6I–K*), and CNU-NX1-742 (*Appendix 1—figure 7A–C*) can be confidently assigned to *Ct. frequens* based on size, wing venation, and colouration, rectangular prothorax and/or long ovipositor.

## 2.2 Ovipositor comparative analysis

This section complements schematic reconstructions provided in *Figure 2*. Schemes representative of Grylloidea, Gryllacrididae, and Anostostomatidae were derived from previous accounts (*Cappe de Baillon, 1920*; *Cappe de Baillon, 1922*; *Kluge, 2016*).

### 2.2.1 *Grylloblatta chandleri* *Kamp, 1963* (schematized under 'Grylloblattodea' in *Figure 3C*)

Our observations corroborate previous accounts (*Walker, 1919*; *Walker, 1943*), in particular regarding the occurrence of a long **olis1** connecting **gp9** and **gp8**. Its **rh** is slightly dejected externally. We also noticed the occurrence of an olistheter interlocking left and right **gp9** along their dorsal margins. A specimen we observed had an egg engaged in the ovipositor. Due to the large diameter of the egg **olis1** unlocked, as well as the dorsal **gp9–gp9** olistheter. It can then be assumed that olistheters are comparatively labile structures in the species. In resting position (i.e. without engaged egg), when viewed externally, the ventral part of **gp9** is not concealed by **gp9**. Most of the area of **gp9** concealed by **gs9** is not as strongly sclerotized as its ventral part, except for the very base and its dorsal, ventral, and apical margins.

### 2.2.2 *Anacridium aegyptium* (*Linnaeus, 1764*) (schematized under 'Caelifera' in *Figure 3C*)

Our observations corroborate previous accounts on other caeliferan species reporting the occurrence of an **olis1** connecting **gp9** and **gp8** along the entire ventral edge of the former (*Kluge, 2016*; *Thompson, 1986*). Unlike reported by *Ander, 1956*, we found no evidence of an olistheter interlocking the 'inner' (i.e. **gp9**) and 'posterior' (i.e. **gs9**) valves (i.e. **olis2**). The **gp8** and *Ander, 1956*, 'lateral basivalvular sclerite' are extensively fused: they share the same lumen, and the dorsal and ventral fusion points are conspicuous in cross-section, owing to a clear invagination, coupled to a substantial and well-delimited thickening, of their shared wall.

### 2.2.3 *Ceuthophilus* sp. (*Figure 3A and B*; schematized under 'Rhaphidophoridae' in *Figure 3C*)

We concur with previous accounts reporting that **olis2** occurs in this lineage and in other Rhaphidophoridae (*Cappe de Baillon, 1920*; *Gurney, 1936*; *Kluge, 2016*). Unlike other orthopterans, the **rh** of **olis2** is a short projection directed posteriorly, while its **al** covers a broader range (as it is, the antero-ventral half of **gp9**). Viewed laterally, the **al** of **olis2** is slightly convex. This configuration possibly provides some degree of rotational freedom to **gs9** vs. **gp9** and **gp8** (interlocked by **olis1**, which extends more posteriorly than **olis2**, including its **rh**), using the **rh** of **olis2** as a slightly movable axis. This supposed ability would allow **gp8** postero-ventral teeth to be exposed (instead of concealed by **gs9**) and then used by the insect to appreciate the adequacy of substrate for oviposition. The **gp8** is only partially concealed by **gs9**.

### 2.2.4 *Tettigonia viridissima* (*Linnaeus, 1758,*) (schematized under 'Tettigoniidae' in *Figure 3C*)

The observed configuration of the ovipositor valves conforms that described by *Cappe de Baillon, 1920*. Unlike assumed by *Kluge, 2016*; among others we argue that the olistheter interlocking **gs9** and **gp8** (thereafter **olis3**) is not homologous with **olis2**. Firstly, a protrusion from **gs9** and directed towards **gp9** (viz., the characteristic features of **olis2**) occurs at various levels along the ovipositor. It is clearly distinct from another well-delimited olistheter (viz. **olis3**). Secondly, as stated by *Kluge, 2016*, the Anostostomatidae possibly represent an 'intermediate' stage is which a well-delimited **olis2** co-occurs with the premises of an **olis3**, in the shape of a projection of the ventral margin of **gs9** into **gp8**. If two olistheters occur (in addition to **olis1**), they cannot be homologous. It follows that there is an **olis3** besides **olis2**.

## 2.3 Analysis of the mandibular MA

Progression of MA curves for the studied taxa are represented in *Appendix 1—figure 9*. Results of the PCA are summarized in *Appendix 1—table 2* and represented in *Appendix 1—figure 10*, including the pPCA. Animated versions of the PCA represented in *Figure 3E* are provided in the associated Dryad dataset (*Chen et al., 2021*).

# 3 Insect species currently known to occur at Xiaheyan

*Palaeoptera*

Rostropalaeoptera

- Palaeodictyoptera
  - ○ *Namuroningxia elegans* **Prokop and Ren, 2007**
  - ○ *Sinodunbaria jarmilae* **Li et al., 2013b**
  - ○ *Xiaheyanella orta* **Fu et al., 2015**
  - ○ *Tytthospilaptera wangae* **Liu et al., 2015**
- Megasecoptermorpha
  - ○ *Brodioptera sinensis* **Pecharová et al., 2015b**
  - ○ *Sinopalaeopteryx splendens* **Pecharová et al., 2015a**
  - ○ *Sinopalaeopteryx olivieri* **Pecharová et al., 2015a**
  - ○ *Namuroptera minuta* **Pecharová et al., 2015a**
  - ○ *Sinodiapha ramosa* **Yang et al., 2020**

Odonatoptera

- *Shenzhousia qilianshanensis* Zhang and Hong, 2006 in **Zhang et al., 2006**
- *Oligotypus huangheensis* **Ren et al., 2009**
- *Tupus orientalis* Zhang, Hong, and Su, 2012 in **Hong et al., 2012**
- *Erasipterella jini* Zhang, Hong & Su, 2012 in **Hong et al., 2012**
- *Aseripterella sinensis* **Li et al., 2013a**
- *Sylphalula laliquei* **Li et al., 2013a**

*Neoptera*

Dictyoptera

- *Qilianiblatta namurensis* **Zhang et al., 2013**
- *Kinklidoblatta youhei* **Wei et al., 2013**
- Undetermined sp.1 (see **Wei et al., 2013**)
- Undetermined sp.2 (see **Wei et al., 2013**)
- Undetermined sp.3 (see **Wei et al., 2013**)

Grylloblattida

- *Sinonamuropteris ningxiaensis* **Peng et al., 2005**

Plecoptera

- *Gulou carpenteri* **Béthoux et al., 2011**

Archaeorthoptera

- *Sinopteron huangheense* **Prokop and Ren, 2007**
- *Chenxiella liuae* **Liu et al., 2009**
- *Longzhua loculata* **Gu et al., 2011**
- *Heterologus duyiwuer* **Béthoux et al., 2012a**
- *Miamia maimai* **Béthoux et al., 2012b**
- *Xixia huban* **Gu et al., 2014**
- *Protomiamia yangi* **Du et al., 2017**
- *Sinogerarus pectinatus* **Gu et al., 2017**
- *Phtanomiamia gui* **Chen et al., 2020**
- *Ctenoptilus frequens* sp. nov. **Chen et al., 2020**

