## [Decision Letter]

**Acceptance summary:**

Using the novel Reflectance Transforming Imaging techniques combined with geometric morphometrics, Chen and colleagues investigated a new lobeattid insect from the Pennsylvanian (Carboniferous) of North China, and demonstrated that the new form possessed a particular mechanism interlocking its elongate ovipositor parts thereby allowed to lay eggs into the ground. The results shed new lights into the debates on Orthoptera, currently representing the bulk of polyneopteran insect diversity, and their early diversification.

**Decision letter after peer review:**

Thank you for resubmitting your work entitled "Ovipositor and mouthparts in a fossil insect support a novel ecological role for early orthopterans in Pennsylvanian forests" for consideration by *eLife*. Your article has been reviewed by 3 reviewers, including Min Zhu as the Reviewing Editor and Reviewer #1, and the evaluation has been by Patricia Wittkopp as the Senior Editor. The following individual involved in the review of your submission has agreed to reveal their identity: Joachim Huag (Reviewer #2).

The manuscript has been improved but there are some remaining issues that need to be addressed, as outlined below:

1) The phylogenetic part should be strengthened to a larger extent, in order to clearly demonstrate the phylogenetic signals of relevant anatomical traits.

2) "Reflectance Transforming Imaging (RTI)" should be explained in the "Methods" section, as it is a new technique to show the anatomical details of insect fossils.

3) Although some explanation for the PCs are given in the text, the factor loadings for each PC should be provided in the supplement. Another file that should be made available, best in the supplement, would be all outlines of the investigated mandibles.

*Reviewer #1:*

Using the novel Reflectance Transforming Imaging techniques combined with geometric morphometrics, Chen and colleagues investigated a new lobeattid insect from the Pennsylvanian (Carboniferous) of North China. They demonstrated that the new form possessed a particular mechanism interlocking its elongate ovipositor parts thereby allowed to lay eggs into the ground. The main claims of the manuscript are supported by the data in overall, and the Reflectance Transforming Imaging techniques used here are informative for broader readers, in addition to entomologists and evolutionary biologists. However, the phylogenetic part of this manuscript should be strengthened to a larger extent, in order to clearly demonstrate the phylogenetic signals of relevant anatomical traits.

Line 37: add "living" in front of "Orthoptera".

Line 39: "abundance" to be clarified. Species and/or individual (specimen) abundance?

Line 48,52: "the earliest known insect faunas" vs "this past insect fauna"?

Line 59: "this abundant fraction of the earliest insect faunas" redundant, replaced by "lobeatid insects"?

Line 72: "Reflectance Transforming Imaging (RTI)" should be explained in the "Methods" section, as it is a new technique to show the anatomical details of insect fossils.

Line 75: "decisive" seems too strong. Anyway it is a statistical result.

Note genus and species names in italics (in several figures).

Line 225: add "crown-group or crown" in front of "Orthoptera".

Line 264-275: The cladogram in Figure 2G is a simplified diagram? Where is the source of this diagram? Without any phylogenetic analysis in this study, it is very difficult to discuss the phylogenetic impacts or signals of anatomical details revealed by the new species, considering the substantial parallelism in insect evolution. This paragraph should be largely revised to corroborate your statements.

Line 333: art credit?

*Reviewer #2:*

In this study, the authors provide very interesting fossils of early orthopteran insects. The images are of very high quality, both photos and drawings. These fossils provide new and very important insights into the early diversification of winged insects. Especially the comparison to extant representatives is well performed, using geometric morphometric methodology for inferring ecological function of the mouthparts of the fossils. With their data, the authors are able to draw conclusions about the phylogenetic position, the feedings habits as well as the oviposition habits of the species.

I have only few suggestions to improve the manuscript, mostly focussing on explaining some aspects for the wider readership and some other minor aspects as, for example, getting rid of ranked taxonomy and providing some more insights into the extant comparison.

Title: As *eLife* is a non-specialists journal, I suggest to add a number for the age of the Pennsylvanian (for me as a biologist this does also not immediately relate to something).

line 29: I do not immediately know what "lobeattid" means. Given the readership of the journal, this needs explanation.

line 49: not sure whether a semantic or philosophical issue, but to my understanding "faunas" are not populated.

line 50: suggest to add some broader groups to the named ones, e.g. "griffenflies" (Odonatoptera) and "megasecopterans" (Palaeodictyopteroidea), again to provide the not-too-deep specialists some frame.

line 56: nothing causes more useless discussion than ranks of groups. I therefore suggest to avoid ranks altogether (there is ample literature on this), in this case the term "order". I think this is in line with the style of the systematic palaeontology section which was also kept largely rankless: I also suggest to get rid of the ranks in the supplementary information.

line 62: in the age of growing creatonism and alike, I would always go for the more careful wording; "ground pattern" is less problematic than "groundplan". This also applies to other instances in the text.

line 70: again "order" could be substituted by a rankless expression such as "lineage".

line 103: the term "gonapophyses" indicates that these structures are of sternal origin. As most researchers instead agree that they are derived parts of appendages, they should be better attributed as "gonopods".

line 331: there were other candidates for endophytic egglaying (e.g. palaeodictyopteroideans), that could be mentioned here.

line 378 "recent" is no longer considered a valid time, "extant" would be the alternative.

Figure legends: Species names should be put in italics.

I would like to see some more of the supplementary figures in the main paper, but I guess that is due to length restrictions of the journal.

Although some explanation for the PCs are given in the text, the factor loadings for each PC should be provided in the supplement. Another file that should be made available, best in the supplement, would be all outlines of the investigated mandibles.

*Reviewer #3:*

Authors investigated large samples of Palaeozoic lobeattids, stem-relatives of all Orthoptera from the Xiaheyan locality using Reflectance Transforming Imaging combined with geometric morphometrics in order to assess lobeattid morphology, infer its ecological role, and phylogenetic position. The analysis of their ovipositor shape indicates that ground was the preferred substrate for eggs and mouthparts indicate omnivory, which explains the paucity of external damage on contemporaneous plants foliage. These very interesting and important results were achieved with use of Reflectance Transforming Imaging (RTI) files for details of several specimens and statistical tools. The results enabled to place analysed fossils in context of their palaeoceological role in the Pennsylvanian habitats but also in phylogenetic context, providing a definitive demonstration that caeliferans are derived from ensiferans. This gives the clue to the debate of evolutionary history and debate on Orthoptera, currently representing the bulk of polyneopteran insect diversity and their early diversification.

---

## [Author Response]

The manuscript has been improved but there are some remaining issues that need to be addressed, as outlined below:1) The phylogenetic part should be strengthened to a larger extent, in order to clearly demonstrate the phylogenetic signals of relevant anatomical traits.

We appreciate the opportunity to be able to place the anatomical traits in a phylogenetic context in more detail in the main text. To address this issue, we deeply reworded the first paragraph of the section ‘Ovipositor morphology and phylogeny’ within the Discussion section and modified the Results section.

Our phylogenetic reasoning is essentially based on the occurrence of a secondary olistheter, which is uniquely present in extant Ensifera. It is therefore clear that the new fossil is either a stem- or a crown-orthopteran. At the same time, the fossil species can be clearly excluded from the taxon Saltatoria because it lacks jumping hind legs. This aspect is further corroborated by the lack of a wing venation feature present in all extant Orthoptera and some of their closest stem-relatives. It is therefore clear that the new fossil species is a stem-orthopteran.

The phylogenetic affinities of the new fossil species are assessed based on an unambiguous combination of inherited and derived character states. Arguably, resorting to taxon names ‘Saltatoria’ and ‘Panorthoptera’ without explanation on their relation with ‘Orthoptera’ made our reasoning confusing. We hope the proposed new wording clarified this aspect.

The item composing our new Figure 3 (formerly Figure 2G) now better illustrates our reasoning for the placement of the new species. The phylogeny is based on a recently published account (Song et al., 2020), which indeed was not referenced adequately (it originally appeared in the section 1.4.2 of the supplement). This is the most comprehensive account on the phylogeny of Orthoptera to date, and the outcome of this account is largely consistent with other recent studies, as far as the taxon subset we are focusing on is concerned.

2) "Reflectance Transforming Imaging (RTI)" should be explained in the "Methods" section, as it is a new technique to show the anatomical details of insect fossils.

We fully agree that this method needs more explanation in order to easen its more widespread use in the palaeontological community. We added more explanations on page 12 of the manuscript within the Methods section which reads as follows:

“RTI files are interactive photographs in the sense that light orientation can be modified at will. The approach, originally developed in the field of archaeology (see Earl et al., 2010 and references therein), has also been applied to a variety of sub-planar fossil items (Béthoux et al., 2016; Hammer et al., 2002; Jäger et al., 2018; Klug et al., 2019; among others).

We computed RTI files based on sets of photographs obtained using a custom-made light dome as described elsewhere (Béthoux et al., 2016), driving a Canon EOS 5D Mark III digital camera coupled to a Canon MP-E 65 mm macro lens. Sets of photographs were optimized for focus using Adobe Photoshop CC 2015.5. RTI computing was then performed using the RTIbuilder software (Cultural Heritage Imaging, San Francisco, CA, USA) using the HSH fitter (a black reflecting hemisphere placed next to the area of interested provided reference). Several snapshots were extracted using the RTIviewer software (Cultural Heritage Imaging, San Francisco, CA, USA), including those in ‘normals visualization’ mode, which provides a color-coded image according to the direction of the normal at each pixel (i.e. the direction of the vector perpendicular to the tangent at each pixel; see Figure 2C and F). This allows to quantify subtle height differences in fossilized structures.”

3) Although some explanation for the PCs are given in the text, the factor loadings for each PC should be provided in the supplement. Another file that should be made available, best in the supplement, would be all outlines of the investigated mandibles.

We fully agree and added the factor loadings in the supplement in table S3. We also provide a new figure S9 in the supplement showing the outlines of all investigated mandibles. In addition we provide a new file “MA_outlines.blend” within the Dryad repository, which can be opened with the open source software Blender (www.blender.org) in order to obtain all coordinates of the MA outlines for reanalysis of the dataset.

Reviewer #1:Using the novel Reflectance Transforming Imaging techniques combined with geometric morphometrics, Chen and colleagues investigated a new lobeattid insect from the Pennsylvanian (Carboniferous) of North China. They demonstrated that the new form possessed a particular mechanism interlocking its elongate ovipositor parts thereby allowed to lay eggs into the ground. The main claims of the manuscript are supported by the data in overall, and the Reflectance Transforming Imaging techniques used here are informative for broader readers, in addition to entomologists and evolutionary biologists. However, the phylogenetic part of this manuscript should be strengthened to a larger extent, in order to clearly demonstrate the phylogenetic signals of relevant anatomical traits.

The main suggestion of this review (improvement of the phylogenetic inference) is addressed above (1^st^ main point).

Line 37: add "living" in front of "Orthoptera".

Changed.

Line 39: "abundance" to be clarified. Species and/or individual (specimen) abundance?

We refer to high individual numbers in this case and clarified the text accordingly.

Line 48,52: "the earliest known insect faunas" vs "this past insect fauna"?

We deleted “past insect” and tried to increase readability of this paragraph by several other modifications.

Line 59: "this abundant fraction of the earliest insect faunas" redundant, replaced by "lobeatid insects"?

Changed accordingly.

Line 72: "Reflectance Transforming Imaging (RTI)" should be explained in the "Methods" section, as it is a new technique to show the anatomical details of insect fossils.

We applied modifications as requested, please also refer to the second main point of the editors mentioned above.

Line 75: "decisive" seems too strong. Anyway it is a statistical result.Note genus and species names in italics (in several figures).

Changed accordingly.

Line 225: add "crown-group or crown" in front of "Orthoptera".

Changed accordingly.

Line 264-275: The cladogram in Figure 2G is a simplified diagram? Where is the source of this diagram? Without any phylogenetic analysis in this study, it is very difficult to discuss the phylogenetic impacts or signals of anatomical details revealed by the new species, considering the substantial parallelism in insect evolution. This paragraph should be largely revised to corroborate your statements.

We applied modifications as requested, please also refer to the first main point of the editors mentioned above. In short, the cladogram is based on the most recent transcriptome based study of Song et al., (2020) and we mapped the character states of the relevant anatomical details on this phylogeny. Larger modifications were applied in the text sections referring to our phylogenetic reasoning.

Line 333: art credit?

Proper credits is now given. We also provide a letter from the author stipulating that we can reproduce this artwork.

Reviewer #2:In this study, the authors provide very interesting fossils of early orthopteran insects. The images are of very high quality, both photos and drawings. These fossils provide new and very important insights into the early diversification of winged insects. Especially the comparison to extant representatives is well performed, using geometric morphometric methodology for inferring ecological function of the mouthparts of the fossils. With their data, the authors are able to draw conclusions about the phylogenetic position, the feedings habits as well as the oviposition habits of the species.

We thank the reviewer for his/her encouraging comments.

I have only few suggestions to improve the manuscript, mostly focussing on explaining some aspects for the wider readership and some other minor aspects as, for example, getting rid of ranked taxonomy and providing some more insights into the extant comparison.Title: As eLife is a non-specialists journal, I suggest to add a number for the age of the Pennsylvanian (for me as a biologist this does also not immediately relate to something)

We added an age in the title and hope this is in line with the journal style.

line 29: I do not immediately know what "lobeattid" means. Given the readership of the journal, this needs explanation.

The sentence was reworded so that the term ‘lobeattid’, arguably cryptic, appears later, in a position where is it better connected with the points being addressed.

line 49: not sure whether a semantic or philosophical issue, but to my understanding "faunas" are not populated.

We changed the wording to “composed”.

line 50: suggest to add some broader groups to the named ones, e.g. "griffenflies" (Odonatoptera) and "megasecopterans" (Palaeodictyopteroidea), again to provide the not-too-deep specialists some frame.

We made modifications accordingly.

line 56: nothing causes more useless discussion than ranks of groups. I therefore suggest to avoid ranks altogether (there is ample literature on this), in this case the term "order". I think this is in line with the style of the systematic palaeontology section which was also kept largely rankless: I also suggest to get rid of the ranks in the supplementary information

We fully agree and rephrased all occurrences accordingly.

line 62: in the age of growing creatonism and alike, I would always go for the more careful wording; "ground pattern" is less problematic than "groundplan". This also applies to other instances in the text.

We changed this accordingly at all occurrences.

line 70: again "order" could be substituted by a rankless expression such as "lineage".

Changed.

line 103: the term "gonapophyses" indicates that these structures are of sternal origin. As most researchers instead agree that they are derived parts of appendages, they should be better attributed as "gonopods".

Changed.

line 331: there were other candidates for endophytic egglaying (e.g. palaeodictyopteroideans), that could be mentioned here.

Agreed and changed accordingly.

line 378 "recent" is no longer considered a valid time, "extant" would be the alternative.

Changed.

Figure legends: Species names should be put in italics.

Changed.

I would like to see some more of the supplementary figures in the main paper, but I guess that is due to length restrictions of the journal.

Relocating some of the supplementary figures to the main text would entail a relocation of the corresponding descriptive sections, which we believe the readers of *eLife* will have limited interest for. Nevertheless, we now divided the original Figure 2 into two, with items E-G being parts of a new figure (now Figure 3). This allowed to add extracts of RTI files of the two fossilized ovipositors to the new version of Figure 2 (items C and F).

Although some explanation for the PCs are given in the text, the factor loadings for each PC should be provided in the supplement. Another file that should be made available, best in the supplement, would be all outlines of the investigated mandibles.

Agreed. We provide now the requested data in the supplement. See also our comment to the respective comment of the editor.

Reviewer #3:Authors investigated large samples of Palaeozoic lobeattids, stem-relatives of all Orthoptera from the Xiaheyan locality using Reflectance Transforming Imaging combined with geometric morphometrics in order to assess lobeattid morphology, infer its ecological role, and phylogenetic position. The analysis of their ovipositor shape indicates that ground was the preferred substrate for eggs and mouthparts indicate omnivory, which explains the paucity of external damage on contemporaneous plants foliage. These very interesting and important results were achieved with use of Reflectance Transforming Imaging (RTI) files for details of several specimens and statistical tools. The results enabled to place analysed fossils in context of their palaeoceological role in the Pennsylvanian habitats but also in phylogenetic context, providing a definitive demonstration that caeliferans are derived from ensiferans. This gives the clue to the debate of evolutionary history and debate on Orthoptera, currently representing the bulk of polyneopteran insect diversity and their early diversification.

We thank the reviewer for his/her assessment of our study.